# Introduction to Quantization of Conformal Gravity

**Lesław Rachwał** [†] 

Departamento de Física-Instituto de Ciências Exatas, Universidade Federal de Juiz de Fora, Juiz de Fora 33036-900, MG, Brazil; leslaw.rachwal@ufjf.edu.br
† Speaker at TQTG'21 ("The Quantum & The Gravity") Conference.

**Abstract:** A method for consistent quantization of conformal gravity treating conformal symmetry in a very controllable way is presented. First, we discuss local conformal symmetry in the framework of gravitational interactions, where we view it as an example of a general gauge theory. We also present some early attempts at quantization of conformal gravity and use the generalized framework of covariant quantization due to Faddeev and Popov. Some salient issues such as the need for conformal gauge-fixing, an issue with conformal third ghosts, and discontinuities in conformal gravity are studied as well. Finally, we provide some explanations of the original ad hoc methods of computation valid at the first quantum loop level in conformal gravity.

**Keywords:** quantum gravity; quantum field theory; higher-derivative gravity; conformal symmetry; conformal gravity; covariant quantization

## 1. Introduction and Motivation

Conformal gravity is a very promising model of relativistic gravitational dynamics. It embodies not only symmetries of general relativity (diffeomorphisms) but also is invariant under conformal transformations of the metric tensor. In this class of theories we view the metric tensor $g_{\mu\nu}$ as the fundamental variable completely describing the gravitational field. In $d = 4$ spacetime dimensions, conformal gravity, (also known as Weyl gravity) is naturally a four-derivative model. This means that from the beginning we have to deal with higher-derivative theories [1,2], contrary to the original Einstein–Hilbert theory which was described just by two-derivative dynamics. Hence, Weyl gravity is the first model of modified gravitational dynamics, and this was introduced by Hermann Weyl in 1918, just a few years later after the original Einstein construction. When describing the dynamics of the gravitational field in a conformal manner we have to be very careful and pay some special attention to this higher-derivative nature of quantum conformal gravity [3,4]. Generally, the higher-derivative nature of gravity is quite inevitable when one considers the effects of quantum matter fields [5]. Conformal gravity is a special case of such a generated (or induced) gravitational theory when all the matter fields are massless.

Conformal gravity is the simplest gravitational model in which we have only dimensionless gravitational couplings. This is why one of the prerequisites for it is scale invariance on the classical level. By placing further restrictions, this latter is constrained more and results in conformal invariance, where the metric tensor is effectively charged under conformal group. This being said, in conformal gravity, we cannot have any standard dimensionful gravitational coupling such as famous gravitational Newton's constant $G_N$ or the cosmological constant. On the classical level, this theory is completely described by the action which is a 4-dimensional volume integral of the conformal Lagrangian. The last one is given by the square of the Weyl tensor (tensor of conformal curvature) according to the formula with contractions, $C^2 = C^2_{\mu\nu\rho\sigma} = C_{\mu\nu\rho\sigma}C^{\mu\nu\rho\sigma}$, where, in the last tensor $C^{\mu\nu\rho\sigma}$, the indices are raised using the contravariant metric tensor $g^{\mu\nu}$. This is the unique four-dimensional Lagrangian, which is conformally covariant (transforms with a conformal weight $w = -4$) in such a way that together with the proper metric density $\sqrt{|g|}$ they

make the Lagrangian density $\sqrt{|g|}C^2$ a conformal invariant exclusively in $d = 4$ spacetime dimensions.

There are various motivations for considering classical conformal gravity as a viable theory of the gravitational field [6–8]. First, the relativistic dynamics of such a system are constrained even more due to the presence of additional symmetry besides the diffeomorphism symmetry. Local conformal symmetry leads classically to new conservation laws and to new integrals of motion. For example, the trace of the energy-momentum tensor $T = g_{\mu\nu}T^{\mu\nu}$ must vanish all the time. This conformal symmetry is also essential for constructing new, exact gravitational solutions. It is also essential for the issue of black holes dynamics, high-energy particle physics [9–11], and also for the issues of singularities [12]. Conformal gravity is a first scale-invariant model; hence, it is consistent with what is generated from quantum loops of matter when the last ones are integrated out on the level of path integral. The gravitational action of conformal gravity can absorb all UV divergences which are produced on non-trivial curved spacetime when massless matter fields exist there. This also implies that conformal gravity in its minimal framework gives rise to a consistent renormalizable theory of gravitational interactions when it is considered in the quantum field theoretical (QFT) framework. On the classical vacuum level, one proves that all Ricci-flat solutions of Einstein gravity are automatically solutions in conformal gravity in four dimensions as well [13]. Finally, the presence of additional symmetry of gravitational interactions is instrumental in solving the issue with spacetime singularities, which are otherwise quite ubiquitous [14–16]. With the power of conformal symmetry, one can show that all curvature singularities are eradicated [15] and that they are not a problem anymore for classical gravitational theory.

Above, we have given mainly theoretical reasons for studies of conformal gravity. There exists also successful phenomenological predictions of conformal gravity which makes this gravitational model completely falsifiable using future gravitational experiments and also observations from astrophysics and cosmology. As a model of modified gravitational dynamics (but still preserving Lorentz symmetry), conformal gravity explains, for example, flat rotation curves in galaxies without the need of local dark matter components. Moreover, when picking the solutions of conformal gravity, one has to pay some special attention to the global (conformal) aspects of spacetime, since the conformal gravitational dynamics fully realize Machian ideas. Additionally, in conformal gravity, one has naturally classical cosmological solutions which describe both early inflationary phase as well as late-time accelerated expansion of our Universe. This is achieved without explicitly adding the cosmological constant term to the action or without a specially designed inflaton scalar field. In the latter case, for exponential evolution, we do not need any dark energy ingredients in the contents of the Universe. To summarize conformal gravity solves the current important problems of present-day cosmology without explicitly invoking dark matter, dark energy, and inflaton fields. For a review of these experimental and observational issues, one can consult [17–19]. One can also study interactions of conformal gravity with particle physics models, especially with its impact on the Higgs potential, Stueckelberg fields, and also on spontaneous breaking of conformal symmetry generated from matter fields such as in [20,21]. For phenomenological applications, it is interesting, for example, to study evaporation of black holes in conformal gravity [22,23], and their computation of entropy [24], the RG flow towards the IR regime [25,26] in one-loop conformal quantum gravity, and also its implications on the dark sectors of the universe as studied in [27].

### 1.1. Motivations and Important Issues on the Quantum Level

With the above theoretical and phenomenological successes of conformal gravity on the classical level, one is tempted to use it also to describe the quantum dynamics of the gravitational field. Here, one enters into the domain of quantum gravity (QG), where the things become much more complicated. For the moment, there is not any experimental proof or evidence that the conformal gravity is a correct theory of quantum gravity. In our framework, adopted in this paper, we want to quantize the dynamics of the gravitational

field in a covariant relativistic manner and in a way that attempts to preserve all classical symmetries of the model. The most successful and suitable approach here is to consider the quantized theory of gravitational interactions in the setup of old good quantum field theories. In this minimal and conservative approach, we study the quantum theory of the gravitational field and its interactions with matter fields. In particular, we focus on the propagation of quanta of gravitational radiation (gravitons), their consistent interactions with other matter particles, and also their self-interactions. The way quantum particles interact is dictated and is very strongly constrained by the symmetries of the quantum theory in question, and here, the fact that we also use conformal symmetry should help in finding a unique and consistent theory of quantum gravitational interactions.

The first results regarding quantum conformal gravity are promising. This model in $d = 4$ spacetime dimensions is renormalizable so there is a control over UV divergences. This already improves over the situation existing in quantum gravity described by the Einstein–Hilbert model in $d = 4$. On the quantum level, one can ensure that gravitons' interactions are consistently described with the diffeomorphism symmetry of any gravitational theory. One generally knows quite well how to quantize the system in such a way that this last local symmetry is completely preserved on the level of quantum dynamics and in the general-relativistic (GR) setup. There exist well-known procedures of covariant quantization due to Faddeev and Popov, and ways to deal with gauge-fixings, gauge conditions, and additional fields of ghostly nature needed only on the quantum level for diffeomorphism local invariance. All these developments can be understood in the framework of quantization of general gauge theories, where the symmetries are local. However, the situation with local conformal symmetry understood in the framework of GR (so equivalently on curved spacetime background) is more intricate.

The quantization of gravitational theories with conformal symmetry faces the problem of the fate of this symmetry on the true quantum level. If one works in the framework of conformal field theories (CFT), then on the quantum level, in order to preserve conformal invariance, very special conditions must be satisfied [28]. The presence of unbroken conformal invariance means that, in particular, all correlation functions must show a scale-invariant behavior and they cannot depend on any mass or energy parameter [29]. This implies that there cannot be any UV divergences in the quantum model. The theory must be completely UV-finite. There are only a few known examples of such theories in the gravitational framework [30–33], and in particular around maximally symmetric spacetime backgrounds [34]. More simple, however, are models with only gauge symmetries on flat spacetime [35]. All the results of loop computations in such models must provide finite convergent results. There should not be any reason for renormalization of the theory and for hiding infinities, hence the scale of arbitrary renormalization $\mu_0$ should be absent in the formulation of the theory on the quantum level. No need for $\mu_0$, no need for renormalization, and only finite redefinitions of couplings are possible in such a model with true quantum conformality. From the point of view of the renormalization group (RG) flow, such a model sits always at the fixed point (FP) so there is actually no RG flow. All the beta functions of couplings must be zero at these circumstances. Due to the absence of any dependence on the energy scale, the quantum fluctuations, corrections, and phenomena, they all look the same, no matter at which scale one looks at them. The situation at the FP of RG, where the scale invariance is automatically realized, allows one to consider full upgrade to the quantum conformal symmetry. Then, one can speak about true quantum CFT, where the conformal symmetry is so powerful to severely constrain correlation functions (for example, 3- and 4-point functions are completely determined), and holding conformal Ward identities places strong conditions on the resulting dynamics of the quantum system. All this is happening obviously when the conformal symmetry is not violated on the quantum level.

When one tries to quantize the pure conformal gravity model $C^2$ in $d = 4$ spacetime dimensions in a natural covariant way, one unfortunately finds that the above conditions do not hold. The theory, for example, at the one-loop level, is not UV-finite; there are

non-vanishing beta functions of dimensionless coupling constants of the theory. There is, however a hope, since not all is lost regarding the conformal symmetry in such circumstances. The counterterms that one needs in this quantum model are also conformally invariant when understood as terms in the gravitational action. Therefore, although the UV divergences are present, then the divergent form of the effective action also *formally* preserves the conformal symmetry. One could say that at this level the conformal invariance is hardly violated

In actuality, there are various ways that the conformal symmetry can be broken on the quantum level. They include violations due to the conformal anomaly (CA) [36–39], a non-trivial RG flow of some dimensionless coupling parameters, a presence of some dimensionful parameter in the quantum theory and its running [40], an addition of some non-conformal deformation parameter, or, finally, via vacuum expectation value of some conformally charged scalar field present in the theory [20]. We would like to strongly emphasize here that despite that in most generic situations the conformal symmetry is present on the classical level, but then disappears on the first quantum level, etc., the quantization procedure is never a formal reason for the violation of conformality. After all, quantization is a mathematical procedure invented by humans, and nature does not have to follow this, nor did it ever, since we know that the quantum Planck constant $\hbar$ is already non-zero. We stress that if the quantum conformal symmetry is broken, then this is because of the effects of some quantum physical phenomena. As remarked above, there are various physical means due to which conformal symmetry may not be fully realized on the quantum level, but we will not discuss them here, leaving this interesting, though difficult, future research.

We stated that the generic situation with Weyl gravity on the quantum level is that at the quantum level the conformal symmetry is only partially realized. At the classical level this symmetry is fully present, then on the first loop level it is violated by the RG running of the dimensionless coupling constant of the theory, i.e., $\beta_{\alpha_C} \neq 0$, where $\alpha_C$ is a scale-dependent coupling in front of the $C^2$ term in the action of the Weyl theory. Still, at the one-loop level, counterterms are conformally invariant and of the form $\sqrt{|g|}C^2$ and $\sqrt{|g|}$GB, where the last one is the famous Gauss–Bonnet topological density. Then, due to the presence of CA already at the one-loop level, the worse situation is expected at the two-loop level. The anomaly heralds the presence of the $R^2$ term in the effective action, which is not generally conformally invariant—it is only invariant with respect to the so-called restricted conformal transformations satisfying the condition $\Box\Omega(x) = 0$. However, in pure Weyl gravity, the situation at the second and higher loops requires more detailed studies. Generally after the quantization, in Weyl gravity, conformal symmetry is not fully visible on the quantum level, but for this drawback, the formal quantization procedure cannot be blamed.

*1.2. A Need for New Conformal Quantization Method*

From the purely theoretical view, one should still have a reliable quantization method that allows for a clean distinction between spurious artifacts and true physical effects which are responsible for violation of quantum conformality. For example, such a method when applied to the FT conformal supergravity model ought to provide results and rules which do not break conformal symmetry, except when this is really needed (but then it is fully under control), such as in the case of performing conformal gauge-fixing. The same method, when used in the case of ordinary conformal gravity (due to Weyl), would give everyone a theoretical advantage of clear separation of sources for violation of conformal invariance on the quantum level. Then, one could better understand the fate of conformal symmetry there and whether this survives quantum corrections. Additionally, such a framework with properly quantized conformal gravitational theory will be a good starting point for writing all covariant Feynman rules of the theory. This, in a long perspective, will allow us to perform perturbative computation of various theoretical processes in quantum conformal gravity to higher loop orders too. It is known, for example, that the covariant technique of

traces of differential operators is not easily applicable to the two-loop level and beyond, while the computation using Feynman diagrams, although very tedious, could be still attempted using some computer software programs that now can deal very effectively with tensor algebra.

In particular, such two-loop explicit results could shed some light on the fate of consistency of pure quantum Weyl gravity since, as we remarked above, the two-loop level is the first one where we should see, perhaps, some disastrous influence of conformal anomaly. Therefore, this computation there could be crucial, and for this we need a very secure way of obtaining Feynman rules of the theory. We need to include the propagators of all fields in the spectrum (also those additional, such as Faddeev–Popov ghost and third ghost fields) and also pay some special attention to perturbative vertices of interactions and the role of proper gauge-fixing of all local symmetries present in the model. The same also regards, for example, the case of the two-loop level accuracy of computation in the FT supergravity theory in $d = 4$. It is expected that at this level one could explicitly derive the relation that must exist between the Yang–Mills and conformal gravitational couplings in this model. The one-loop condition for UV-finiteness of the full quantum coupled model is not sensitive to such relation; however, the presence of it is needed for a consistent absence of perturbative divergences at higher loop orders and also for support of some claims of even more extended supersymmetry there, which would be, presumably, here realized nonlinearly on fields of the theory.

In order to write with confidence such Feynman rules and perturbative spectrum of all propagating modes, one needs to have a very clear and methodological means of quantization of general gravitational theories with conformal symmetry present in the local version. Generally, we think that such a scheme of quantization which treats the conformal symmetry very gently is naturally needed to write and deal with perturbative rules of the models, in which it is known that conformal symmetry is present also in the quantum domain of the theory. Then, by achieving this, one could provide a necessary tool of the preferred quantization method which is consistent with all the symmetries of the model, present both on the classical and quantum level. Such a quantization method does not violate conformal symmetry accidentally, and if it does so, this is performed in a very controllable way. The consistent formulation of such a quantization method within the framework of general gauge theories (with local symmetries) and with various other constraints is possible and we devote the main part of this paper to providing an introduction to such a topic.

We think that, above, we have sufficiently motivated the need for a new quantization method when dealing with local conformal symmetry in the gravitational framework. The quantization method which conserves conformal symmetry is a natural generalization of previous methods which were suitably constructed in such ways to preserve gauge symmetries (in YM theories), diffeomorphisms (in general gravitational theories), and also local supersymmetry (in supergravitational models). We think that this generalization to treat the conformal symmetry consistently is a natural, and probably an ultimate, generalization of all these above methods. This claim could be supported by the belief that the local conformal symmetry in the framework of gravitational interactions is probably the last gauge symmetry to be discovered since it requires a limiting attempt to reach formally infinitely high energies. As we explained above, the true quantum conformal symmetry should probably reside there in the deep UV regime, so then it could be very difficult to be proven or rejected by experiments in the QG domain.

*1.3. Contents of the Article*

The structure of this paper is as follows. First, we discuss the conformal symmetry as a gauge symmetry. This we study in the framework of flat spacetime, curved spacetime (GR), and also in the setup of general gauge theories. In the next section, we present the main details of the construction of the suitable quantization method that preserves conformal symmetry. For this purpose, we initially analyze all the necessary elements of the

construction present already when the theories with diffeomorphisms are quantized using generalized Faddeev–Popov trick. Then, we motivate the need for conformal gauge-fixing and conformal third-ghosts fields. To end this section, we also present a small account of Veltman-like discontinuities in conformal gravity on the classical level and also on the problems with counting perturbative degrees of freedom in the spectrum of Weyl gravity in $d = 4$. Finally, we discuss the pioneering, although a bit ad hoc, method of Fradkin and Tseytlin to arrive at the correct one-loop quantum results in conformal gravity. At the end, we draw our conclusions and sketch some possible future directions of developments of these interesting conformal topics.

## 2. Conformal Symmetry as a Gauge Symmetry in Gravity

### 2.1. Conformal Symmetry in Flat Spacetime

First, one can consider conformal symmetry in the flat spacetime framework. In this setting, one can also generate gravity as a gauge theory of small tensorial fluctuations around flat spacetime. Moreover, conformal transformations may also have their origin here. Originally, we start with scale transformations or global conformal transformations. When one picks up a Cartesian coordinate system $(x^\mu)$ to describe events on flat Minkowski spacetime, the following transformations,

$$x^\mu \to (x^\mu)' = x'^\mu = \lambda x^\mu \tag{1}$$

are called scale transformations for $\lambda > 0$ and $\lambda = $ const. They change how the scales (units) of lengths and distances are measured and defined. By rules of quantum physics, the scale of length corresponds to inverse of the energy scale (the transcription is performed using the Planck constant $\hbar$), so we also effectively change characteristic scales of the energy of the physical system under consideration. The transformation (1) leads to

$$ds^2 \to ds'^2 = \lambda^2 ds^2, \tag{2}$$

where the Minkowskian infinitesimal length element (or proper time $d\tau^2$ along geodesics) $ds^2$ is shown *not* to be invariant under scale transformations. Instead, it rescales with the positive factor $\lambda^2$, so its sign is untouched and this also implies invariance of the character of the world-lines in relativistic physics (time-like, light-like, or space-like). This informs us that the causal structure is not modified by scale transformations, at least on flat spacetime background. It is obvious that for any non-zero $\lambda$, the square $\lambda^2$ is positive in (2), while for definiteness we have chosen $\lambda > 0$ in (1). The infinitesimal distance $ds^2$ being only covariant under scale transformations is fully consistent with finite lengths and energies, also transforming with some scale factors under changes of the scale.

Already in the flat spacetime field theory, a requirement of scale invariance (so, invariance of the physical observables of a model under scale transformations (1)) is quite constraining. Namely, in the construction of such a classical field theory we cannot obtain help from any dimensionful quantity such as an energy scale or a length scale (in high energy units, where $[E] = [L]^{-1} = $ GeV). In addition, all coupling constants of the model describing interactions have to be properly dimensionless when the dimension of the underlying Minkowski spacetime is fixed. In particular, this condition excludes any presence of mass parameters for fields in the model. The fields themselves, of course, may carry energy dimensions according to their canonical (engineering) dimensionalities. As a result, we end up with a model which, for its definition, does not rely upon any particular choice of physical units of mass, time, length, energy, etc. This is contrary to other models that, for their definitions, have to use ratios of some scales with some other reference scales. Additionally, one could say that in such a theory there is no scale (scale-less or scale-free models). On the quantum level, full quantum scale invariance implies that there is no renormalization group (RG) flow of all the necessarily dimensionless couplings of the theory, so the theory sits all the time at the fixed point (FP) of RG flow.

Still, on flat spacetime, which in four dimensions comes with the full Poincaré symmetry group $SO(1,3) \ltimes T^4$, scale symmetry can be viewed just as a one-dimensional group of global transformations isomorphic to a positive ray of the real line: $\mathbb{R}_+$. The subgroup of rescalings (also known as conformal dilatations) directly multiplies the full Poincaré group since it is known that a commutator of two conformal rescalings does not additionally boost, rotate, or translate the physical system. This is not the case with the relation between Lorentz subgroup $SO(1,3)$ and translation subgroup $T^4$, and they enter into Poincaré group being multiplied semi-directly. If one focuses on the 10-element Poincaré group, then one can map it into $SO(2,3)$ group with translations $T^4$ included as "rotations" in extended five-dimensional spacetime. This results in a non-trivial structure of the algebra of commutators between generators (both boosts and rotations) from the full Poincaré group.

In almost all classical field theory examples usually considered on flat spacetime, the scale invariance can be promoted to a bigger symmetry group, which includes also "vectorial"-like transformations (or "translations"-like transformations) [28,41]. This group, together with Poincaré factor, constitutes a 15-dimensional conformal group Conf on four-dimensional flat spacetime. The five additional generators compared to the group containing standard spacetime symmetries (Poincaré algebra has 10 generators) are one dilatation (for scale symmetry) and a four-dimensional vector of special conformal transformations (SCTs). One can see that the full conformal group Conf is isomorphic to $SO(2,4)$, so to the group related to Lorentz transformations but only in a six-dimensional spacetime with signature $(+,+,-,-,-,-)$. This could be understood by following the so called "embedding formalism" for conformal symmetry due to Dirac. The structure of the full 15-element conformal group Conf is more complicated (than just of the underlying Poincaré group $SO(1,3) \ltimes T^4$) due to various semi-direct products used in its definition. Moreover, it is known that two subsequent SCTs also generate resulting Poincaré transformations on a physical system, and that their generators behave similar to Lorentz vectors under boosts and rotations. Therefore, the structure of commutators in the Conf algebra is more intricate around flat spacetime, but we will not need an explicit form for it here.

Finally, we mention that while dilatations can be presented as a global part of the full conformal group Conf and they can be performed with constant transformation parameters $\lambda$, the intrinsic nature of SCTs prevents the same identification for them. The character of SCTs is that they depend on some special Lorentz vector (usually denoted by $b_\mu$), and in an explicit transformation law in four-dimensional Cartesian coordinates, one can see also a non-trivial dependence on the spacetime location point and the Minkowskian distance from the origin $x^2 = x^\mu x_\mu$. This is not so surprising when one recalls that the full conformal group Conf of flat spacetime contains conformal inversion transformations, that is $x^2 \to \frac{1}{x^2}$. To complicate this case, such dependence is found in the denominators of the nonlinear transformation laws for Cartesian four-dimensional standard coordinates[1]. The conclusion is that the SCTs cannot be performed with constant parameters and their action actually depends on the spacetime point characterized by Cartesian coordinates $x^\mu$. This means that, for example, two SCTs with the same vector $b_\mu$ will act differently in two different spacetime points $x_1^\mu$ and $x_2^\mu$ with $x_1^2 \neq x_2^2$. We can say without too much rigor that part of the full conformal group which generates SCTs is already in the "local" version, contrary to the other parts which are global (as dilatations are). This implies that it is actually incorrect to try gauging the generators of SCTs by force [42,43]. One can also convince oneself about this fact by comparing a count of number of generators in the full 15-dimensional conformal group with the number of associated vector gauge bosons. In the conformal part of this group (consisting of dilatations and SCTs) after gauging and making the gauge potentials dynamical, one finds only one gauge boson (Weyl vector field, originally associated incorrectly by Weyl to a photon) corresponding to the generator of dilatations. The SCTs do not add new fields in the gauge theory framework.

One remark is in order here. When we consider matter field theories around flat spacetime background, which are without dimensionful coupling parameters, thus, scale-

invariant and later easily promoted to conformally invariant in the sense of transformations given in (1), and from the full Conf group, then we mean global scale transformations, with parameters of conformal transformations which are *not* spacetime-dependent functions of coordinates $x^\mu$. As explained in the introductory sections, this is the story on the classical level. On quantum level to require full conformality, one must have vanishing beta functions for all dimensionless couplings of the theory, or in other disguised scale invariance of all Green functions, when the fields on external legs are not rescaled. The set of examples for such a behavior is, of course, more restricting, the $\mathcal{N}=4$ super-Yang–Mills (SYM) theory being the standard one. However, we remark that, despite that $\mathcal{N}=4$ SYM on flat background is completely UV-divergence free and scale-invariant, this model of QFT is not invariant under gauged conformal transformations; for example, it is not invariant under conformal dilatation transformations with $\lambda=\lambda(x)$, where the latter is an arbitrary function of spacetime points. Hence, gauging of the conformal group Conf (only parts related to dilatation generator) brings a new kind of symmetry to the system and constrains the dynamics even more. This symmetry with new added values can be fully understood only in the proper gravitational context.

The class of theories (matter models) which are symmetric under transformations from the global and full conformal group Conf (in particular, with respect to dilatations with constant parameters $\lambda=$ const) is already very special and the theories in it are called conformal field theories (CFTs in short). Such CFTs describe quite constrained quantum dynamical matter systems different than ordinary QFTs because here there is no RG flow. The CFT is located at the quantum FP of the RG. The requirement of global (rigid) conformal symmetry places severe constraints on the dynamics, and, for example, quantum (to all orders) two- and three-point functions are completely determined. Higher $n$-point functions satisfy some recurrence relations, but all the story so far has been without any influence of gravity and for global scale (or conformal) transformations with $\lambda=$ const. When one performs the gauging of dilatations and the final theory is brought to the form invariant under local scale transformations with $\lambda=\lambda(x)$, then this framework is already more constraining than the original CFTs were. Similarly, the original CFTs are not global conformal anomaly-free when they are coupled to external background metric of spacetime different than Minkowski metric $\eta_{\mu\nu}$[2]. Therefore, coupling to external metric field (to gravity) or gauging the conformal group Conf takes us out from the domain of CFTs analyzed around flat background and we must necessarily consider conformal symmetry in the gravitational setup. Here, there are various ways possible to perform the mentioned above gauging [44], such as Ricci gauging, Weyl gauging, etc. For the moment, only very special coupled theories (matter+gravity) satisfy demands of local quantum conformal symmetry, and the participation of quantum conformal gravity in this dynamics is essential.

The analysis of dimensions of various subgroups and the structure of the full conformal group Conf can be extended to be valid in any dimension of spacetime $d$. For this purpose, one only notes that if there is one time dimension (to have a consistent physical description) and $(d-1)$ spatial dimensions, then the Lorentz group is $SO(1, d-1)$ with $\frac{d(d-1)}{2}$ generators, and the Poincaré group is $SO(1, d-1) \ltimes T^d$ isomorphic to $SO(2, d-1)$ with $\frac{d(d+1)}{2}$ generators. Finally, adding one $d$-dimensional vector of SCTs and one generator of dilatations, we obtain the full conformal group Conf isomorphic to $SO(2, d)$ with $\frac{(d+1)(d+2)}{2}$ generators. Of course, these general considerations make sense only for dimensions $d > 2$ since it is well known that in $d = 2$ special dimensions of spacetime (or two-dimensional Euclidean space), the conformal group and algebra are infinite. This infinitely enhanced power of two-dimensional conformal symmetry has its roots in powerful complex analysis on Argand plane, as it is well known and used in mathematics. Moreover, this powerful infinite-dimensional conformal algebra is the reason for various successes of $d = 2$ worldsheet formulation (both Euclidean and Minkowskian) of string theory. Eventually, this gauged conformal symmetry in string worldsheets is so omnipotent that it cancels target spacetime ultraviolet divergences otherwise ubiquitous and usually problematic in other quantum gravity models.

In order to obtain a gauged theory of gravity, one can perform gauging of all 10 Poincaré generators as well as of the dilatation generator from the full conformal groups [45]. The SCTs do not need to be gauged. When all symmetries are made local from global and a special care is exerted to construct physical actions, which are invariant under these new local transformation symmetries, then one can conclude that an extended gauge theory of conformal gravity is accomplished. This is in the framework of gauged Poincaré gravity (with curvature and torsion as two independent field strengths) and with an addition of conformal symmetry [46]. For this end, one only gauges dilatations from the conformal part of the full conformal group Conf. In what follows, we will concentrate only on this conformal part of the whole story. Actually, one can show that first, it is consistent to consider only gauging of the conformal part, leaving untouched the gravitational part of the full 15-element Conf group, and second, for the last end, only dilatation generator and scale transformations (1) need to be changed from global to local (with spacetime dependent parameters $\lambda \to \lambda(x)$ in full generality in (1)).

*2.2. Conformal Symmetry in Curved Spacetime*

Instead of discussing the explicit gauging of the dilatation generator, we will follow below a different, in a sense more geometrical, route. Namely, we will consider conformal transformations and conformal symmetry in the framework brought about by differential geometry of curved manifold backgrounds. That is, in the spacetime setting, we discuss conformal symmetry in a fully general-relativistic (GR) framework [47,48]. By this we do not mean a full dynamical setup coming with Einstein's gravitational theory. For what matters here we will consider only the kinematical aspects of gravitational theory while the specification of precise dynamical content in some models to be considered later will be clear when we will discuss the quantization of dynamical gravitational systems also enjoying conformal symmetry. Therefore, by GR we mean here a theory describing spacetime physics independently of the particular coordinate system chosen and all ensuing consequences of such a view on physical systems.

In actuality, the origin of the construction of the word "conformal" for conformal transformations, maps, images, and geometry is related to the way to preserve a specific geometric form of some objects studied in geometry. In a rough translation from Italian, "con forme" means in English "with the forms" (to be understood with appendix "pre-served"), so conformal means "with the forms preserved". The "forms" in question are angles between vectors which are unchanged in this kind of geometry[3].

In conformal geometry of spacetime, (hyperbolic) angles between four vectors are preserved, and the causal structure (also named because of this conformal structure) remains untouched when one performs conformal geometric transformations. Such a transformation does modify lengths or magnitudes of vectors and scalar products between them. This means that the differential structure on the manifold is not invariant in conformal geometry and depends on the choice of conformal gauge. In addition, the distances (in the sense of four-dimensional proper Minkowskian intervals) are now not absolute, not invariant, and become relative. They all depend on the conformal gauge. The last one is realized by a selection of the conformal factor of the metric (such as $\lambda^2(x)$ in (2) or $\Omega^2$ in (3) below) since, typically, the conformal symmetry does not come with its own set of coordinates, and borrows the ones also used standardly in differential geometry, nor does it often come with its own set of gauge potential fields to be fixed by this choice (the notable exception here is the gauge theory proposed by Weyl with Weyl gauge connections). In the framework of conformal geometry and field theories enjoying conformal symmetry, only angles are still absolute, but they always constitute very beautiful geometrical structures (exemplified in Escher's pictures of anti-de-Sitter-like geometries).

In the GR setup (on a general curved manifold), the scale transformation is the following active operation on the covariant metric tensor field $g_{\mu\nu} = g_{\mu\nu}(x)$ on the manifold

$$g_{\mu\nu} \to g'_{\mu\nu} = \Omega^2 g_{\mu\nu} = e^{2\ln\Omega} g_{\mu\nu}, \tag{3}$$

where $\Omega$ is a constant parameter of the transformation.

This very much resembles a global $U(1)$ transformation of QED for a charged scalar field $\phi = \phi(x)$ given by

$$\phi \to \phi' = e^{i \ln \alpha} \phi, \tag{4}$$

where we also require that $\alpha = $ const. One notices that there is a difference of imaginary unit $i$ between exponents in formulas (3) and (4), respectively. Similarly to the QED case, the metric field $g_{\mu\nu}$ can be treated as being effectively charged under conformal group. Or, even more, the metric fields are matter fields from the point of view of conformal symmetry, they transform similar to electrons but with the real, not a complex, prefactor phase, and the module of the conformal coupling charge of the metric is universal and equals two (by convention).

In general gauge theory, the fields can be divided into two groups: gauge potentials or matter fields. The difference is only how they transform under gauge transformations. Let us assume that these transformations are linear (the fields in question must for this furnish a linear representation of a gauge group) and for definiteness we only consider them in an infinitesimal form. Then, if the transformation laws are homogeneous, then the fields upon which the transformation is performed are matter fields, if instead they are inhomogeneous but still linear, then these are gauge transformations of gauge potentials. We remark that for different symmetries the same field can play different roles, similar to the metric tensor for diffeomorphisms, and GCT is a gauge potential but it behaves similar to matter field for conformal symmetry.

However, there are also very important differences between these two cases, giving rise to the dichotomy between gauge theories (in internal spaces) as used extensively in non-gravitational field theory models and conformal gravitational theories (in external spacetime) used sometimes to describe relativistic gravitational fields. First, for the complex-valued scalar field $\phi$ in (4), the invariant module square, that is $|\phi|^2 = \phi^\dagger \phi$, is QED-invariant, while for the real-valued metric in (3), the Lorentz-invariant infinitesimal length element $ds^2$ is not invariant under scale transformations (cf. (2))[4].

Moreover, due to the fact that in the QED case, the complex phase factor $e^{i \ln \alpha}$ is present, the $U(1)$ group is topologically identical to the circle $S^1$ and hence is compact. In the case of the real and positive scale factor $\Omega^2$ in (3), this is isomorphic to $\mathbb{R}_+$ and hence this is a non-compact group of dilatations. This implies that some mathematical aspects will be very different for the $\mathbb{R}_+$ group of conformal dilatations and the $U(1)$ group of complex phase transformations in QED. For example, when one tries to construct a gauge theory of a scalar field $\phi$ which is invariant under real rescaling of the field, then one cannot use the invariant $|\phi|^2 = \phi^\dagger \phi$ or even simple square $\phi^2$ (such as for the real-valued Higgs field) in construction of the invariant action functional, and even the globally symmetric $\mathbb{R}_+$ theory on Minkowski background is not well-defined. To achieve this, one has to leave the flat absolute Minkowski background and move to the framework of general relativity, where these scale transformations show the structure of spacetime under the influence of gravitational field embodied in the proper metric tensor field $g_{\mu\nu}(x)$.

On the other hand, if one has overcome the above problems, and tries to build a locally symmetric theory, then the differences are present, too. In the case of QED, it is well established that the proper covariant derivative of the scalar field is inherently complex-valued, when the $U(1)$ gauge connection field is taken as real, but this part of the definition for the $U(1)$ gauge covariant derivative must be with the imaginary unit $i$. On the contrary, in the case of group of dilatations, the proper covariant derivative (Weyl covariant derivative with a real Weyl gauge connection field) is entirely real-valued and there is no room for any imaginary unit. Eventually, one should recall the London brothers' correspondence, thanks to which the complex-valued electron fields $\psi$ (analogs of used here charged matter scalar field $\phi$) give interpretation to complex-valued and normalizable electron wave functions in quantum mechanics. Of course, such interpretation is not possible when one deals with real-valued metric fields $g_{\mu\nu}$ which are charged under conformal symmetry group, but their transformation law is with the real positive overall

factor $\Omega^2$, so one stays all the time in the real space formalism, which cannot be applied directly to quantum theory. Additionally, the original identification by Weyl of the Weyl gauge potential as the vector potential of the electromagnetic field suffers from all the above problems and cannot be retained anymore in field theory models. We can understand that in his time Weyl was seeking for a very economical model combining (or even unifying based on some geometrical arguments) together gravitation and electromagnetism. As we think nowadays, he was not right and the road towards a full unification is still very long, because gravitation (even if the models of conformal gravitation are being considered) is quite different from internal space gauge theories (such as electromagnetism and non-Abelian Yang–Mills (YM) theories). However, as we will show below, the formal procedure of gauging the global symmetries in these two models works very similarly and we do not see any big differences here.

By exploiting the well-known formula

$$ds^2 = g_{\mu\nu}dx^\mu dx^\nu, \tag{5}$$

one sees that (3) implies (2)

$$ds^2 \to ds'^2 = \Omega^2 ds^2, \tag{6}$$

in accordance with our previous considerations of conformal transformations on flat space-time background described in Cartesian coordinates. For this, in the GR framework, contrary to (1), one postulates that coordinates $x^\mu$, in any coordinate system, are not changed by the transformations, the same for their contravariant differentials $dx^\mu$.

Gauging means that we promote the constant parameters of transformations to space-time dependent parameters of the same kind of transformations, namely,

$$\alpha \to \alpha = \alpha(x) \tag{7}$$

in QED and

$$\Omega \to \Omega = \Omega(x) \tag{8}$$

in spacetime physics. This last operation produces full-fledged conformal transformations within the meaning of GR:

$$g_{\mu\nu} \to g'_{\mu\nu} = \Omega^2(x)g_{\mu\nu} = e^{2\ln\Omega(x)}g_{\mu\nu} \tag{9}$$

with the local parameter of conformal transformations $\Omega = \Omega(x)$ and there is no need of further gauging of these transformations. This is local scale invariance or conformal invariance in GR. This is, in a sense, a gauging of global scale invariance (3) understood in the GR framework. For general conformal transformations, the parameters $\Omega = \Omega(x)$ are local functions of the spacetime point $x$. We emphasize that within the GR, the conformal transformations are already in the local version and the conformal symmetry could be viewed here as a gauge symmetry of the system, provided that the description of the con-figurations of the system is invariant under action of these transformations. This, of course, depends on the dynamics in the gravitational sector of the theory. We note that, not for all theories, the conformal symmetry in the version from (9) is a symmetry of gravitational dynamics. In actuality, the requirement for such a symmetry of the gravitational action is very restrictive in gravitational models, also when they are coupled to conformal matter models. Basically, when one fixes the number of spacetime dimensions, the number of possible different theories enjoying conformal symmetry, even on the classical level, is finite and is usually quite small. When one moves to lower degenerate dimensions, such as $d = 2$, such theories may not exist at all. Therefore, the requirement for consistent conformal symmetry of the gravitational system constrains quite tightly the possible form of the dynamics in this system.

The local conformal symmetry finds its proper place in the context of gravitational physics, where it is understood as a possible transformation of the spacetime, thus of the

gravitational background. Despite that it was originally studied and constructed on flat absolute Minkowski spacetime and promoted there from simple scale invariance to the full 15-element conformal group (in $d = 4$), its natural embedding is in the setup of differential geometry. This is because a conformal transformation really touches the measurement of distances, so crucial to the mathematical definition of infinitesimal (differential) geometry. Since gravitational field has as one of its manifestations the form of curved spacetime as differential manifold, then the conformal symmetry, or the conformal group in general, will also primarily be related to the dynamics of gravitational fields. At the end, we conclude that the conformal symmetry is one of the possible symmetries of relativistic gravitational fields. This symmetry in a gauged form can be an additional (and last to be discovered) gauge symmetry of gravitation in addition to the local Poincaré group of spacetime symmetries.

We also emphasize that, strictly speaking, the examples of CFTs that we considered in the previous subsection on the flat background (such as $\mathcal{N} = 4$ SYM theory), which are conformal on the full quantum level, are not invariant under scale transformations in GR, even when they are described by constant parameters $\Omega = $ const. In the last case, one could come back easily to the form of Minkowski metric (absolute, not rescaled) by performing a compensating rescaling of Cartesian coordinates according to (1) with $\lambda = \Omega^{-1}$. On flat background, Minkowski metric $\eta_{\mu\nu}$ is considered as an absolute element, which is in obvious contradiction with GR, but even if we allow this compensating coordinate transformation, then this extended setup cannot work on more general manifolds than the flat one. Or, speaking physically, when we couple a flat spacetime CFT to external gravitational field, then such a theory will not be automatically invariant under local (gauged, so with $\Omega = \Omega(x)$ and not rigid anymore) scale transformations with the sense from GR, from (3), and (9) below. Moreover, we know that after such transformations the Minkowski metric $\eta_{\mu\nu}$ changes into a general conformally flat metric of spacetime and not into a necessarily flat one. We remind the reader that the original CFT was not defined in such circumstances. This disappointing feature is also related to the fact that most of flat spacetime CFTs are not free from the conformal anomaly problem in external metric fields.

It must be noted that gauging of dilatation transformations is not the same as appending SCTs to global scale transformation group (isomorphic to $\mathbb{R}_+$) and subsequent promotion of global scale invariance to full global conformal invariance under transformations from the full 15-dimensional Conf group. In the flat spacetime framework presented in the previous subsection, the conformal group Conf was still a group of global transformations, even if it contained somewhat local parts related to SCTs. The situation with gauging of scale transformations (9) on the metric tensor as understood in GR leads to a different group of conformal transformations. These last transformations should be understood only in the framework of GR, and after the gauging is finished, their general form is with precisely one spacetime dependent parameter $\Omega(x)$. Thus, the resulting group of transformations is different than a 15-dimensional global Conf group. The relations between these groups and their flat spacetime global analogs is the same as between a diffeomorphism group Diff and the global Poincaré group $SO(1, 3) \ltimes T^4$ on flat spacetime. The former is known to be infinite-dimensional (because it contains local transformations), while the latter has only 10 generators. Similarly, a local conformal group in GR is infinite-dimensional, while there are one generator of dilatations, four of SCTs, or altogether 15 generators in global flat spacetime Conf group. Hence, these conformal groups are clearly different.

We also remark that although naturally placed in the gravitational and differential geometry context, the conformal symmetry is not the same as a symmetry pertaining to general coordinate transformations (GCTs). For them, we have

$$x \to x' = x'(x), \tag{10}$$

where $x'(x)$ are differentiable but arbitrary functions. Moreover, general coordinate transformations preserve the value of the most fundamental scalar invariant of differential

geometry, namely of $ds^2$. In actuality, this is not a consequence, but rather a postulate from which one can derive, for example, the transformation law under GCTs of the covariant metric tensor field $g_{\mu\nu}(x)$. Taking a quick look at (6), one easily understands that general conformal transformations in GR are, most of the time, not general coordinate transformations. This is, for example, clear from the fact that the group of conformal transformations is different than the group of GCTs, known as diffeomorphisms (and denoted below by Diff). Therefore, in the most general situation in the dynamics of gravitational fields, we can have two symmetries of diffeomorphisms and of conformal transformations. These two symmetries do not overlap each other and can be treated quite separately, for example, for the quantization aims. In a rough sense, one can state that in such a case, the total group is a direct product of subgroups of diffeomorphisms Diff and of a local conformal symmetry (this is a different group than 15-element Conf group, it is, rather, only a group of symmetries under local conformal transformations, so it is, in a sense, a one-dimensional group of gauged dilatations). The two groups do not disturb each other, although they use the same field variables as the objects upon which corresponding transformations act. This is a peculiar feature of metric gravitational theories, where the metric tensor is the basis for everything related to gravitation. These two groups both act on the metric; their actions are, however, distinguished by the character of transformations and by different parameter(s) used to characterize such transformations.

### 2.3. Conformal Symmetry in General Gauge Theories

As emphasized above, the local conformal symmetry in the GR framework can be understood as a local gauge symmetry of the gravitational system, possibly also coupled to some matter models. Therefore, for this symmetry one can apply the general formalism of general gauge theories and their ensuing Faddeev–Popov quantizations. Placing the conformal symmetry in this formalism is particularly convenient for the sake of covariant quantization approach [3,4,49] applied to the classical system of gravitational field, enjoying full conformal symmetry with the hope of obtaining consistent theory of conformal and gravitational interactions. To this end, one first needs to recall some facts about conformal symmetry in this framework. First, we will use a general metric background characterized by the symmetric rank-2 tensor field $g_{\mu\nu}(x)$. Below, and in later sections, we will use the condensed notation due to DeWitt. One can also consult the textbooks [50,51], where this notation was employed for the first time for the case of general gauge theories. One can consider the possibility of obtaining conformal gravity from general gauge theory as in [52].

The conformal symmetry transformations were defined in (9) but for the sake of emphasizing the spacetime dependence, we write the last formula as

$$g_{\mu\nu}(x) \to g_{\mu\nu}(x)' = \Omega(x)^2 g_{\mu\nu}(x), \tag{11}$$

where $\Omega = \Omega(x)$ is a finite parameter of conformal transformations, possibly also spacetime-dependent. In the infinitesimal version, this transformation reads,

$$\delta g_{\mu\nu} = \delta g_{\mu\nu}(x) = \xi(x)g_{\mu\nu}(x) = \xi g_{\mu\nu}, \tag{12}$$

where the infinitesimal parameter of transformation $\xi \ll 1$ is a general spacetime-dependent function $\xi(x)$ and we have that $\Omega^2 \approx 1 + \xi$. Of course, as obvious from (12), the infinitesimal conformal transformations are characterized by only one scalar transformation parameter $\xi$, which is spacetime-dependent for local conformal symmetry (compared with the case of diffeomorphisms, where a GCT is characterized by $d$ parameters collected in one GR-covariant vector $\xi_\rho$). For the purpose of covariant quantization, it is very important to derive the form of an infinitesimal generator $R$ of conformal transformations[5].

The general formula for $R$ reads,

$$R = R(x, y) = \frac{\delta\phi(x)}{\delta\xi(y)}. \tag{13}$$

In our case, when we look for the generator of infinitesimal conformal transformations, the "charged" field is the metric field, so we take $\delta\phi(x) = \delta g_{\mu\nu}(x)$, and the parameter of this transformation is $\xi = \xi(y)$. Therefore, we obtain that in our case, the corresponding formula is

$$R = R_{\mu\nu}(x,y) = g_{\mu\nu}\delta^{(d)}(x,y). \tag{14}$$

As we see, the general form of this generator $R$ inherits two covariant indices from the metric tensor. Moreover, it also possesses ultra-local dependence on two spacetime points $x$ (where the charged field is located) and $y$ (where the parameter $\xi(y)$ resides), and this dependence is expressed via the $d$-dimensional Dirac delta function. In general, one could say that the generator in this case is a bitensor, or more precisely, a rank-2 symmetric tensor (such as the metric tensor $g_{\mu\nu}$) from the point of view of the point $x$, and a scalar from the point of view of the point $y$ [50,51].

Already, here, one can notice a few important things which will be important in what follows later regarding the quantization in different choices of conformal gauges. First, the conformal gauge transformations act on a metric field in a linear way. This is true not only for the infinitesimal version of these transformations, but also for finite local conformal rescalings. This is a good feature simplifying the covariant quantization procedure. The generators create a linear algebra of conformal gauge symmetries, which closes off-shell (and then we do not have to employ the Batalin–Vilkovisky more complicated quantization method). Secondly, the conformal transformations do not involve any derivatives of the metric tensor which are charged here. This is also a feature which is well received, since this will have some positive consequences on the ghost sector of conformal symmetry. In actuality, in the quantization, we can exploit both of them. With the explicit formula in (14), we can conclude the part with understanding of conformal symmetry as a gauge symmetry in the gravitational setup. All further developments depend on the choice of the conformal gauge for quantization procedure, and hence they are not universal pertaining to the knowledge of general gauge theories.

*2.4. Fradkin–Tseytlin Conformal Supergravity*

We must also remark here that there exists an extended model of conformal gravity which is UV-finite on the full quantum level. This is realized when additional matter fields are included but they are all united with gravity in the same multiplet. Moreover, the symmetry which achieves this also mixes bosonic and fermionic degrees of freedom, hence we have the right to call it extended supersymmetry. In a model when this extension is maximal possible and where in the pure bosonic spin-2 sector we deal with Weyl gravity in $d = 4$, we may have a special situation where all perturbative and also non-perturbative divergences completely vanish. Such an extended supergravitational model was first proposed by Fradkin and Tseytlin in 1985 [53]. These two copies of $\mathcal{N} = 4$ super-Yang–Mills theories are coupled to $\mathcal{N} = 4$ conformal supergravity, which is a supersymmetric generalization of the Weyl conformal gravity in the spin-2 sector of fluctuations.

In the $\mathcal{N} = 4$ Fradkin–Tseytlin (FT) conformal supergravity model, the UV divergences were checked for the complete absence at the first perturbative loop level. However, the argument for UV-finiteness in this case is much stronger since the presence of quantum conformal symmetry is guaranteed on the quantum level from some algebraic considerations, and this holds to all loop orders and also non-perturbatively. Basically, being in the same multiplet of currents, the conservation of conformal (virial) current is related by the extended $\mathcal{N} = 4$ supersymmetry argument to the conservation of the matter energy–momentum tensor $\nabla_\mu T^{\mu\nu} = 0$ together with the conservation of vectorial Yang–Mills (YM) current $D_\mu j^{\mu,a} = 0$. From the last statement about the conformal current, one derives the implication of non-violated conformal symmetry in the model purely from algebraic reasons, so it does not matter whether this is on the classical or on the quantum level. Such conservation holds automatically in the theory.

The example of FT supergravity shows that it is possible to find a QFT theory (actually a CFT) which also includes quantum gravitational interactions that are completely free

of any UV infinities and also of classical singularities of spacetime, since the conformal symmetry and invariance are present both on the classical and quantum levels of this model. From the point of view of RG, this is a very special and highly symmetric theory which is defined at the FP of RG, and the CFT which describes it is supersymmetric and with gravity included. It was very difficult to find similar CFT models with gravitation since other known examples considered only matter theories and on flat spacetime backgrounds (such as mentioned before, $\mathcal{N} = 4$ super-Yang–Mills theory [54]). On the quantum level, the finiteness also implies complete absence of conformal (trace) anomaly, and even the trace of the quantum energy–momentum tensor is vanishing, i.e., $T = 0$. The conformal symmetry is fully realized on the quantum level and around any spacetime background in this model. There is not any problem with its violation, and moreover, the power of conformal symmetry is at use here since all Green functions are heavily constrained, similar to in any CFT. This is properly an anomaly-free gravitational theory [55]. Gravitational conformal Ward identities hold and this is the true representation of the fact that conformal symmetry is unbroken here.

One could also wonder whether in such symmetric theories, which are also quantum conformal, there is a further enhancement of supersymmetries beyond the case of $\mathcal{N} = 4$. However, here, the counting of possible generators (charges) of the superconformal algebra excludes such possibilities—at least that such a hypothetical gravitational CFT with higher than 16 supercharges cannot exist around flat spacetime and when all these symmetries are realized linearly [56]. Maybe some nonlinear versions can exist with superconformal algebras with $\mathcal{N} = 8$ unifying together conformal supergravity and also quantum conformal $\mathcal{N} = 4$ super-Yang–Mills theories. For such a model of quantum conformal gravitational interactions, from the physical standpoint we shall view it emerging as true and UV-finite, and thus a completely sensible theory of QG interactions at the very high energies (in the deep UV regime). Therefore, one expects that when one lowers energy scale from the UV limit, the conformal invariance is somehow broken and we end up with effective models not so symmetric and without full constraining power of all symmetries of the gravitational theory. One should relate this, in the future low-energy phenomena, with the breaking of conformal symmetry. Moreover, one should be able to embed in this gravitational framework without UV-divergences some UV-finite models of particle physics, as were proposed in [57,58].

### 3. An Example of Covariant Quantization of Diffeomorphisms

At the beginning, we remind the successful covariant quantization prescription when applied to a generic higher-derivative gravitational theory. As emphasized in Section 2, gravitational theories with conformal symmetry, away from dimensionality of spacetime being two, $d > 2$, are necessarily very special types of higher-derivative gravitational theories, and with only all couplings dimensionless. The general higher-derivative (HD) theories were quantized in the covariant way by applying the general Faddeev–Popov (FP) methods based on functional analysis (originally such covariant quantization scheme, but only working up to one-loop level, was invented by Feynman). In the general gravitational setup, FP analysis has to be exploited for diffeomorphism symmetries, which are the gauge symmetries of gravitation (at least in the minimal setting). The consideration of higher-derivative theories of gravity, rather than Einsteinian gravity, is here beneficial, as we will see shortly. The situation with higher derivatives is the most general possible one, more general than the case of two-derivative gravitational theories. This will also be important for quantization of conformal symmetry in various gauges and in various dimensions of spacetime.

The expression for the true quantum partition function of a general gravitational theory with diffeomorphisms reads,

$$Z_{\text{grav}}[J^{\mu\nu}] = \int \mathcal{D}h_{\mu\nu}\mathcal{D}\bar{C}^{\alpha}\mathcal{D}C_{\beta}\mathcal{D}b_{\gamma} \exp\left[i\left(S_{\text{grav}} + S_{\text{gfix}} + S_{\text{gh}} + h_{\mu\nu}J^{\mu\nu}\right)\right], \qquad (15)$$

where the main quantum field is the metric fluctuation field $h_{\mu\nu}$ defined as a deviation from the background metric $g_{\mu\nu}$ such that the total metric is $g_{\mu\nu} + h_{\mu\nu}$. Notice that we have decided that our main quantum field will be a symmetric rank-2 tensor field with two covariant indices—this is our choice of the parametrization of the quantum metric field. As stated before, we work exclusively in metric theories of gravity and we do not allow any dynamical or quantum variables with torsion or non-metricity of spacetime. Therefore, the choice of the covariant fluctuations $h_{\mu\nu}$ as the quantum variable is very natural. The covariant metric tensor field $g_{\mu\nu}$ is the most standard choice since the coordinates are typically chosen as contravariant objects $x^\mu$ and this metric tensor enters into the expression (5) to construct a Diff invariant scalar, that is, $ds^2$. Of course, other choices of the parametrization of quantum metric fluctuation variable are also possible. Moreover, although not explicitly mentioned, here we work in general background field method, keeping the background classical metric $g_{\mu\nu}$ as fully general. In this way, we realize all physical requirements related to hyped background independence of quantum gravity. Because of this, the functional of quantum partition function $Z_{\text{grav}}$ in (15) also contains not-path-integrated background metric $g_{\mu\nu}$.

In (15), we path-integrate (notation with $\mathcal{D}\phi$ for differentials of fields under path integrals) not only over the quantum metric fluctuation field $h_{\mu\nu}$, but also over a set of additional quantum fields needed because of the diffeomorphism symmetry present in the model and because of a general higher-derivative nature of gravitational theory. All fields in this set are good ghost fields needed to keep the BRST remnants of diffeomorphism symmetry also on the quantum level of the theory and to avoid potential problems with anomalies. They are added to make the quantization procedure covariant and to preserve diffeomorphism symmetry, such that the theory after the procedure (quantum theory) is defined as with fully realized BRST symmetry originating from the full classical Diff symmetry, the same as the original classical theory defined by $S_{\text{grav}}$. The quantization performed in this way controls very well the local symmetries of the model, and there is no any spurious or accidental breaking of diffeomorphism or any other local symmetry. The framework fixes a coordinate gauge (of diffeomorphisms) by adding the action $S_{\text{gfix}}$, but in a very controllable way. One can fully follow the dependence on gauge-fixing parameters present in $S_{\text{gfix}}$ and convince oneself that in the final physical results the dependence on them completely drops out. In this way, the quantization act does not violate the local symmetries, here diffeomorphism symmetries. This is the essence of the meaning of covariant quantization procedure. The quantum theory obtained in this way is gauge-fixed but with gauge-fixing performed in a very clear and parametric way. The BRST symmetry is what remains at the quantum level after gauge-fixing of all the local symmetries. It controls the form of the total action $S_{\text{tot}}$ appearing in the exponent of the integrand of path integral in (15),

$$S_{\text{tot}} = S_{\text{grav}} + S_{\text{gfix}} + S_{\text{gh}}. \tag{16}$$

### 3.1. Ghost Fields

The ghost fields needed for the local gauge consistency of the model are of two types here. The original FP ghosts are complex-valued quantum fields (the first ghost $C_\beta$ and the second ghost $\bar{C}^\alpha$ being the anti-ghost of $C_\beta$), and a third ghost real field $b_\gamma$. Since the local symmetries here are diffeomorphisms, then each of these ghosts come with one Lorentz index, so they have intrinsically vectorial nature from the spacetime point of view. We emphasize that the canonical position of these spacetime indices is covariant index on the FP ghost field $C_\beta$, contravariant index position on the FP anti-ghost field $\bar{C}^\alpha$, and covariant index on the third ghost $b_\gamma$. Since the covariant metric tensor of fluctuations $h_{\mu\nu}$ is our quantum variable, then it is also natural to choose a linear diffeomorphism gauge-fixing condition as a covariant vector $\chi_\mu$. Then, the fact that a parameter of infinitesimal Diff transformation is also taken as a covariant vector $\xi_\nu$ (because $\delta g_{\mu\nu} = \nabla_\mu \xi_\nu + \nabla_\nu \xi_\mu$) implies that the ghost field has to be with one covariant index $C_\beta$ in the canonical position, since in the derivation of the FP determinant this ghost field takes position of an arbitrary

parameter $\xi_\nu$. The anti-ghost field has to contract in a natural way with the covariant index on the gauge-fixing condition $\chi_\mu$ from the left side, so it must carry a contravariant index according to $\bar{C}^\alpha$, as we wrote above. Finally, the third ghost field substitutes the diffeomorphism gauge-fixing $\chi_\mu$, hence its index is also naturally covariant in $b_\gamma$.

All these ghosts are spurious fields appearing only on the quantum level since they have anti-commuting character and mathematically are described using Grassmannian variables (similar to fermions), although they do not obey the usual spin-statistics theorems. They never appear on external legs of any correlation functions and also never in the perturbative spectrum of the theory, hence they are harmless for the unitarity issue. However, they are crucial for the issue of gauge invariance of the resulting effective quantum theory. Since they were not present in the classical theory and they show up only on the quantum level when one uses the covariant quantization prescription, then the reason for their names is obvious. They are classically non-existing, anti-commuting fields, therefore they satisfy classical anti-commutation relations,

$$\left\{ \bar{C}^\alpha, \bar{C}^\kappa \right\} = \left\{ \bar{C}^\alpha, C_\beta \right\} = \left\{ \bar{C}^\alpha, b_\gamma \right\} = \left\{ C_\beta, C_\lambda \right\} = \left\{ C_\beta, b_\gamma \right\} = \left\{ b_\gamma, b_\varkappa \right\} = 0 \qquad (17)$$

even before these fields are treated as quantum operators in the second quantization (not needed here). These "phantom" fields are present only inside perturbative loops of the quantum covariant theory and they are needed there, for example, to secure the gauge invariance of the final results of computation of some Green functions or physical processes using Feynman diagrams. The necessity to introduce them as virtual particles running in the loops is the price to pay when one wants to deal with gauge symmetry and redundancy of the formalism of general gauge theories on the quantum level in a perturbative and covariant manner. These ghost particles never appear as asymptotic states or on external legs (that is, on-shell, satisfying classical equations of motion (EOM) of the theory) of any Feynman diagram. In actuality, it is without any sense to speak about classical EOM for ghosts or on-shell conditions for ghost fields since the classical action and ensuing EOM for physical fields are defined entirely by the functional $S_{\mathrm{grav}}\left[ g_{\mu\nu} \right]$.

FP ghosts are not visible on the level of classical theory described by the action $S_{\mathrm{grav}}$, where the gauge-fixing procedure is not necessary to obtain classical equations of motion for the classical metric field $g_{\mu\nu}$. They are not present in the perturbative spectrum of classical theory (obtained from classical two-point functions when the background is chosen and some gauge-fixing has to be performed); hence, these fields have nothing to do with the violation of unitarity in higher-derivative theories[6]. If one moves to the level of classical higher $n$-point functions (again when the background is specified and gauge-fixing is performed), one also does not see any effect of these truly quantum virtual modes. Therefore, we draw a clear distinction between these good ghosts and bad ghosts present in higher-derivative theories and related to the famous Ostrogradsky theorem. The latter ones arise at the perturbative spectrum of classical theory, and in principle can appear on external legs before they are not consistently forbidden to appear there by various prescriptions of how to save perturbative unitarity in classical higher-derivative and non-local theories [60–62].

Let us here discuss in more detail this issue with Ostrogradsky instabilities on the classical level or equivalently with the presence of "bad ghosts" on the tree level of the spectrum of the quantum theory. Generally, in any class of higher-derivative theories (including not only gravity but also other fundamental interactions), perturbative unitarity of $S$-matrix is in danger because there are perturbative states with negative kinetic energy or equivalently with negative mass square parameter—so-called ghosts or tachyonic states, respectively [63]. In fact, the optical theorem implies that the $S$-matrix based on the perturbation theory in such cases is non-unitary. Obviously, both ghosts and tachyons are undesirable, and various approaches have been invoked to remove them (or their effects) from the observable predictions of the theory.

One of such approaches was based on the consideration of higher-than-quadratic polynomials (in derivatives or curvatures) in the gravitational actions. However, soon it

was realized that with every polynomial theory, the ghost states must inevitably appear in the physical spectrum of particles [64]. Besides this, various cures have been proposed in the literature for dealing with the ghosts–tachyon issue: Lee–Wick prescription [65,66], fakeons [67–69], non-perturbative numerical methods [48,70–72], ghost instabilities [73,74], non- Hermitian *PT*-symmetric quantum mechanics [75,76], when applied to conformal gravity [6–8,77], etc. (see also other discussions in [62,78]). One might even entertain the idea that unitarity in quantum gravity is not a fundamental concept [79–81]. We think that so far, none of the proposed solutions conclusively solves the problem. However, we also express the hope that these various research directions on the issue of unitarity in conformal gravity will be revealed to have a lot in common. Maybe in the future they will merge into one, conclusively and definitely solving this problem since in each of them one finds a bunch of good theoretical ideas.

These last bad ghosts show up also in other disguises on the level of exact and perturbative classical solutions [82,83], where they signal instabilities of the theory in a form of various possible runaway solutions [73,84]. Some progress with resolution of these important problems can be seen in [85]. The clearest way to see the presence of Ostrogradsky ghosts (also called Boulware–Deser ghosts in higher-derivative gravitational theory, or simply Weyl ghost in Weyl conformal four-dimensional gravity) is to study the form of the perturbative tree-level propagator of the theory, for example, around flat spacetime background and to perform a simple fraction decomposition there. Bad perturbative ghosts are defined as modes whose residues are negative in this last decomposition of the propagator of all fluctuating modes of the theory. They are dangerous for the unitarity of the resulting quantum theory but various ingenious proposals were invented to circumvent these dangerous (or even disastrous for the whole consistency of the theory) effects of them. The FP ghosts are truly different.

Since FP ghosts are complex-valued, in order to give a real total action $S_{\text{tot}}$, they always have to show up in a complex pair, which we take as $\bar{C}C$, as is usually performed with complex fields (such as fermions or charged scalar fields). The remaining third ghost field $b_\gamma$ is real and this condition does not apply; however, as one sees, the essence of the FP construction of introducing anti-commuting ghost fields to the quantum theory is to rewrite Faddeev–Popov determinants as expressions quadratic in the corresponding ghost fields, so we also necessarily have a pair of third ghost fields in the ghost actions below. Namely, we have

$$S_{\text{gh}} = \int d^d x \sqrt{|g|} \left( \bar{C}^\alpha M_\alpha{}^\beta C_\beta + \frac{1}{2} b_\gamma G^{\gamma\delta} b_\delta \right). \tag{18}$$

As is clear from the construction of this action $S_{\text{gh}}$, the ghost fields do not have higher self-interactions, and they do not interact between themselves either. They do interact only with the background metric $g_{\mu\nu}$, actually to infinite order in the last field. The requirement of this construction for quadratic and diagonal action (between FP sector and the sector of the third ghost) originates from the desire to rewrite various positive powers of determinants of various differential operators as integrations over some quantum fields under path integrals. Since these powers are positive, one has to use Grassmannian field variables to achieve such expansion. As is well known for this purpose, one can consistently concentrate on the expressions which are quadratic in these new ghost variables. For simplicity, higher terms with self-interactions of ghosts are not included in this minimal FP construction. In actuality, the Grassmannian nature of ghosts forces them to appear every time in the action with even powers (similar to standard fermions), and the power two is here the minimal possible choice.

The numerical factors in (18) correspond to standard normalization of the Lagrangians for canonically normalized scalar fields which are complex (similar to FP ghosts) and real (similar to the third ghost). The two matrix-differential operators $M_\alpha{}^\beta$ and $G^{\gamma\delta}$ (with the canonical positions of spacetime indices as indicated) can be understood as kinetic operators (defined on a general background) between respective ghost fields. They give rise to the dynamics of the ghosts on the quantum level (inside perturbative loops) and that is why

the quantum propagator for these ghost fields exists and is invertible. In conclusion, one can judiciously write Feynman rules for the ghosts, in particular, the rules for ghost lines (ghost propagators), and use them in construction of more complicated Feynman diagrams of the theory. One notices that the two propagators are completely independent and there is no mixing between FP ghosts and the third ghost field. Moreover, the precise forms of these matrix-differential operators $M_\alpha{}^\beta$ and $G^{\gamma\delta}$ can be found in general higher-derivative gravitational theories, but they are not relevant in the present discussion.

Simply speaking, the requirement to preserve the BRST remnant of gauge invariance in the quantum theory forces, due to the presence of the FP determinant, the introduction of FP two ghosts fields $\bar{C}^\alpha$ and $C_\beta$, while the third ghost field $b_\gamma$ is introduced only when one deals with higher-derivative theories (possessing higher than two-derivative classical EOM). In actuality, in the case of two-derivative theories, one can still deal with the third ghosts and their kinetic operator $G^{\gamma\delta}$ (also called gauge-weighting operator), but this is a superfluous construction. The simpler one is just to take $G^{\gamma\delta}$ in the form proportional to the contravariant metric function $g^{\gamma\delta}$ multiplied by ultralocal $d$-dimensional Dirac delta function in two spacetime points $x$ and $y$, without any differential operator character. Then, in such a situation, the sector of the third ghost is completely algebraic, since their action $S_{3\mathrm{gh}}$ reduces to $b^2 = b^\gamma b_\gamma$ and then they do not have any dynamics, their propagator does not exist, and they do not show up in perturbative loops. Then, simply, one can safely forget about them on the quantum level and for all covariant quantization purposes.

The third ghosts are instead helpful and are a convenient trick in higher-derivative theories, especially when considered on general background or in background field method formalism (so also in pure non-Abelian gauge theories). Their role is related to the third and last part of addition in $S_{\mathrm{tot}}$ in (16), namely, to the action functional denoted as $S_{\mathrm{gfix}}$.

*3.2. Gauge-Fixing*

Now, we come to the discussion of this last element of the total covariant action used in the Faddeev–Popov procedure for quantization. The last part of the action in (16) is entirely responsible for adding the gauge-fixing in a covariant manner to the gauge-invariant action, which for gravity was denoted as $S_{\mathrm{grav}}$. Fixing of the gauge is a necessary step in the covariant quantization procedure due to Faddeev and Popov. Typically, one achieves this by exploiting some gauge-fixing conditions $\chi$ particularized to various local symmetry groups present in the model. In the case of quantum gravity, these conditions are standardly taken as linear local conditions in metric fluctuation fields $h_{\mu\nu}$. Due to the character of a coordinate choice related to a specific gauge choice, these gauge-fixing conditions should be covariant vectors from the spacetime point of view $\chi_\gamma$ and they should carry precisely one covariant index. Their Lorentz form is exactly the same as for the third ghost fields $b_\gamma$. The precise form of the linear gauge-fixing conditions $\chi_\gamma$ is not important for the discussion that follows. We just remark that because of Lorentz covariance, they should contain odd number of derivatives acting on the metric fluctuation field $h_{\mu\nu}$ and, of course, in the smallest version, this means precisely one covariant derivative (constructed with the usage of the background metric $g_{\mu\nu}$) acting on one fluctuation field $h_{\mu\nu}$ (the last is due to the assumed linearity of the gauge-fixing condition $\chi_\gamma$ in $h_{\mu\nu}$). However, of course, some forms of $\chi_\gamma$ can be selected from some broad families of gauge-fixings and then these conditions $\chi_\gamma$ contain a dependence on various gauge-fixing parameters, usually called $\beta$, $\gamma$, etc.

In the minimal version, applicable with full success to two-derivative theories, the gauge-fixing action $S_{\mathrm{gfix}}$ is enough to be constructed from the square of the gauge fixing condition $\chi^2 = \chi^\gamma \chi_\gamma$ properly weighted by another gauge-fixing parameter (which is often called $\alpha$), hence the name of the gauge-weighting functional $G^{\gamma\delta}$; here, it is equal to $\frac{1}{2}\alpha^{-1}g^{\gamma\delta}$. We need to remark that, in general, the gauge-fixing functional $G^{\gamma\delta}$ should not be confused with the gauge-fixing condition $\chi_\gamma$. In actuality, on the classical level, to solve EOM one typically chooses a coordinate gauge for diffeomorphisms by a condition $\chi_\gamma = \ell_\gamma$, where $\ell_\gamma$ is some covariant vector field on the manifold; in particular, this last one can be selected to be zero. What we do instead, on the level of action, is fix the gauge by conditions which

are quadratic in $\chi_\gamma$, which, of course, are not gauge-invariant conditions. By varying these conditions on the level of action with respect to the metric fluctuation field $h_{\mu\nu}$, one obtains related linear gauge-fixing conditions, but still different than original $\chi_\gamma$ (because they often contain more derivatives), to be used during solving EOM or finding higher $n$-point functions of classical theory. Moreover, by employing the gauge-fixing functional $G^{\gamma\delta}$, we effectively perform an average over all conditions $\ell_\gamma$ according to 't Hooft ideas, when, again, $G^{\gamma\delta}$ plays the role of the weighting functional during this mathematical averaging process.

In a more general case, of theories with higher derivatives, the general form of the gauge-fixing action is given by

$$S_{\text{gfix}} = \int d^d x \sqrt{|g|}\, \frac{1}{2\alpha} \chi_\gamma G^{\gamma\delta} \chi_\delta, \tag{19}$$

where the matrix-differential operator $G^{\gamma\delta}$ is quite arbitrary (besides concurring with Lorentz symmetry in its two contravariant indices $\gamma$ and $\delta$, but it does not even have to be symmetric as an operator in these two indices). The choice of the particular form of this operator $G^{\gamma\delta}$ is dictated solely by practical purposes. Namely, one also wants to be able to invert the kinetic operator between (gravitational) fluctuations on a general background, in particular around flat spacetime. This choice of $G^{\gamma\delta}$ is parameterized by various gauge-fixing parameters, such as $\alpha$, $\beta$, etc.

The situation around flat spacetime for metric fluctuations is that in two-derivative Einsteinian theory, gauge-fixing achieved by addition of $\frac{1}{2\alpha}\chi^2$ is completely sufficient, and the Hessian (kinetic operator between fluctuations) is completely invertible as an operator. This happens because, typically, the gauge conditions $\chi_\gamma$ already contain one derivative acting on metric fluctuations $h_{\mu\nu}$ and, after integrating by parts, one sees that from the term $\frac{1}{2\alpha}\chi^2$ one obtains precisely an addition to the kinetic term of graviton with exactly two derivatives. This is sufficient to completely remove the degeneracy of the kinetic operator between small gravitational perturbations and to find the propagator (being the inverse of the Hessian) by algebraic inversion procedure.

The general feature of theories with local symmetry is that the kinetic operator (Hessian, or second variational derivative with respect to fluctuating gauge modes) is degenerate as a consequence of local symmetries present. To remove this problem and make the operator invertible, one needs to add some gauge-fixing terms which explicitly break gauge symmetries, but are still under full control due to the presence of controlling parameters, such as $\alpha$, $\beta$, etc. Only then can the perturbative propagator of gauge modes be found, even around flat and flat gauge connection backgrounds. Typically, such propagator then explicitly depends on gauge-fixing parameters $\alpha$, $\beta$, etc., but here this is not a problem since the propagator by itself (or even when embodied as a two-point function in quantum theory) is not an observable, physical quantity. The gauge-fixing functional $S_{\text{gfix}}$ is then crucial even for finding flat spacetime propagator.

Now, one can easily understand the nuisances of higher-derivative theories (both gravitational or pure gauge theories). In such circumstances, the kinetic operator is with higher derivatives, and addition of just a term $\frac{1}{2\alpha}\chi^2$ containing always only two derivatives, does not modify the degeneracy due to local gauge symmetry on the level of the Hessian and on the level with the highest number of derivatives, which is then, here, higher than two. One solves the problem by adding the full non-minimal form of the gauge-fixing functional as presented in (19). One also has to require that the matrix operator $G^{\gamma\delta}$ now contains some derivatives and cannot be purely algebraic and proportional to $g^{\gamma\delta}$. Of course, the precise number of derivatives in it depend on the number of higher derivatives above two present in the construction of classical EOM of the theory.

In general, the matrix-differential operator $G^{\gamma\delta}$ is not algebraic and must be chosen as a self-adjoint operator for a consistent construction of the quantum theory, or theory of linear perturbations to a quadratic level, so on the level of the effective action to the accuracy of one loop. Exactly the same operator $G^{\gamma\delta}$ appears in the kinetic action for the

third ghost fields. This is not a mere coincidence but a fact stemming from the consistency of the whole covariant quantization formalism in gauge theories. Hence, exactly the same considerations can be repeated for the dynamics in the sector of third ghost fields. One now understands that they are crucial for perturbative covariant treatment of gauge theories with higher derivatives in the classical action.

As clearly seen from the action in (18) and resulting from the perturbative vertices, the third ghosts interact only with background metric (or gauge) fields. This is already a great simplification for their treatment in the method of Feynman diagrams. For example, when restricted to the flat spacetime perturbation theory, third ghosts may not couple to metric fluctuations $h_{\mu\nu}$, and the background is trivial $\eta_{\mu\nu}$, so despite that they have their own propagator and internal dynamics in the sector, one can completely forget about their effects in all perturbative loops. For this, one chooses the form of matrix-differential operator $G^{\gamma\delta}$ as proportional to the metric $\eta^{\gamma\delta}$ and constructed out of exclusively flat spacetime d'Alembertian operator $\partial^2$. Of course, such a choice does remove the degeneracy in kinetic operator for gravitons around flat spacetime and allows the perturbative propagator to be computed. However, from some esthetical reasons, one can object that the form of the operator $G^{\gamma\delta}$ when proportional to powers of $\partial^2$ is in disagreement with general covariance of the framework used for gravitational theories. Of course, this is the price for not having effects of third ghost fields around flat spacetime.

One could instead use a generally covariant operator $G^{\gamma\delta}$ constructed now out of $\Box = \nabla^2 = g^{\mu\nu}\nabla_\mu\nabla_\nu$, but then the interactions between perturbative gravitons $h_{\mu\nu}$ and third ghost fields $b_\gamma$ in the second part of (18) would be resurrected, even on flat spacetime background. However, even on flat background when using a non-covariantly-looking form of the matrix-differential operator $G^{\gamma\delta}$ built with $\partial^2$, one can forget about the third ghosts in Feynman diagrams, but one cannot forget about adding the gauge-fixing term $\chi_\gamma G^{\gamma\delta}\chi_\delta$ to the action of the theory to fix the degeneracy and to be able to find the perturbative propagator.

When using the theory in other backgrounds than flat, the interactions of third ghost fields $b_\gamma$ with gravitons $h_{\mu\nu}$ (perturbations around this chosen background) are naturally generated. Therefore, the presence of the third ghost field is very natural in higher-derivative gauge theories when analyzed using background field method on general curved backgrounds and in covariant manner. Moreover, one notices that the similar arguments as raised above about the simplification happening on flat spacetime background for the third ghost field cannot simplify the sector of complex normal FP ghosts. This is because the matrix-differential operator $M_\alpha{}^\beta$ cannot be constructed only using partial derivatives on flat spacetime $\partial_\mu$ or their square in the form of the d'Alembertian operator $\partial^2$. The construction of the FP ghosts kinetic operator $M_\alpha{}^\beta$ requires that at least one derivative there must be fully gravitationally (or gauge) covariant. That is, in the framework of perturbative quantum gravity, this must be a generally covariant derivative $\nabla_\mu$ since such is used in the infinitesimal generator for local symmetry of diffeomorphisms. All this is a simple consequence of the fact that the FP operator $M_\alpha{}^\beta$ is the *covariant* change of the gauge-fixing condition $\chi_\gamma$ under infinitesimal coordinate transformations (diffeomorphisms in gravity or gauge transformations in full generality). Of course, now the formalism looks much better when all derivatives used in the operator $M_\alpha{}^\beta$ are generally covariant, but this, of course, implies non-trivial form of interaction between FP ghost fields and perturbative gravitons around any background and up to infinite order in graviton fields.

### 3.3. Gravitational Action and Correlation Functions

Having discussed at length the issues related to gauge-fixing and ghost actions and to their construction, now we come to the last ingredients present in the expression for the general partition function in (15) in general higher-derivative gravitational theories.

This simplest element is the gravitational action with higher derivatives $S_{\text{grav}}$. In a general setting, when we work within background field method, this functional

$$S_{\text{grav}} = S_{\text{grav}}[\hat{g}_{\mu\nu}] = S_{\text{grav}}[g_{\mu\nu}, h_{\mu\nu}] \tag{20}$$

depends on both the fluctuation (graviton) fields $h_{\mu\nu}$ (over which we path-integrate) as well as on the background metric $g_{\mu\nu}$. Only in the special case of flat spacetime is this entirely a functional of only graviton fields: $S_{\text{grav}}[h_{\mu\nu}]$. In the general situation, this action, when understood classically, is with all local symmetries realized on the level of full metric tensor $\hat{g}_{\mu\nu} = g_{\mu\nu} + h_{\mu\nu}$, hence it must be written generally covariantly with respect to the full metric $\hat{g}_{\mu\nu}$. The construction of such actions with higher derivatives is a standard topic that we will not repeat here. As we emphasized in previous sections, we will deal with actions that, on top of local symmetries related to diffeomorphism group, also possess the invariance under local conformal transformations understood in the proper GR framework. As explained earlier, the construction of such actions is more tricky and the results are more constrained.

Above, we have discussed the issues of how to build parts of the total action in (16) responsible for gauge-fixing, for fully consistent ghost sector(s), and also for the main gravitational part. This discussion will be very useful and very beneficial when we will deal in a strikingly similar manner with quantization of local conformal symmetry in the next section. The form of analysis of presence or algebraic triviality of various sectors, various fields, or various operators for local conformal symmetry will parallel much of what we have just said in this section. There are many common similarities between quantization of diffeomorphism and local conformal symmetry, since both of them fit in the broad framework of general gauge theories. We will see that the situation with all additional quantum sectors present in $S_{\text{tot}}$ in (16), as it is in general HD gravitational theories, for local conformal symmetry will find its counterparts in some models of conformal gravity in some dimensions and with some specific conformal gauge choices.

Finally, we remark that in the form of the general partition function in (15), we added in the exponent of the integrand a coupling of graviton fields $h_{\mu\nu}$ to an external current (could be viewed as an external energy–momentum tensor (EMT) of some external matter fields) $J^{\mu\nu}$ preserving both locality and general covariance of the extended action. This is a known trick in quantum field theory since all Green functions are now, in principle, obtainable by respective variational differentiation of the full partition function $Z_{\text{true}}[J^{\mu\nu}]$ with respect to the current field $J^{\mu\nu}(x)$ and then setting its vanishing value. We add that in general background field method, the partition function in (15) depends also functionally on the background metric $g_{\mu\nu}$ which was not path-integrated out in this formula, hence it is obvious that the full non-perturbative partition function $Z_{\text{grav}}$ is, in a sense, dependent on the choice of the classical background.

Moreover, another emanation of ghostly nature of fields $\bar{C}^{\alpha}$, $C_{\beta}$, and $b_{\gamma}$ is that they do not come with any classical background values (although some generalizations in this direction are possible [50]) and we path-integrate only over their, in a sense, quantum fluctuation fields. These fields possess only quantum loop nature, as explained above. This means that from our form of the partition function in (15) we cannot, even in principle, obtain correlation functions with any of these ghost fields on external lines, because we can differentiate only with respect to the current $J^{\mu\nu}$ naturally coupled to the graviton field $h_{\mu\nu}$. Therefore, within this view it is senseless to speak about Green functions with good ghosts on external legs, or even about scattering amplitudes with ghosts on external lines, since it is without sense to place them on-shell since they do not come with any well-defined classical EOM. However, some generalizations in these directions are also possible, and a quantum theory with correlators between ghosts and gauge fields can be also formulated, and for some aspects these are even fruitful extensions.

Additionally, the Green functions one obtains from (15) when the external current $J^{\mu\nu}$ is turned on are, as it is well known, not properly gauge-invariant. They would be in the absence of this current, but then from the partition function we could obtain only

one sensible amplitude, namely, the vacuum persistence amplitude or the transition of perturbative vacuum into vacuum. Of course, such an amplitude has its meaning in gravitational theory, *per se*, although it is typical in usual quantum field theory around flat spacetime or in statistical mechanics to treat this one number as just a convenient normalization of other Green functions. Along with standard knowledge here, we know that the presence of external current violates gauge invariance of the full system of quantum correlators. However, the resolution comes from the fact that the physical observables are typically taken not as rough correlators, but as *S*-matrix elements between dressed and on-shell fields derived from general *n*-point functions. For such objects, not for mere quantum correlation functions, Weinberg's theorems guarantee that physical observables are gauge-invariant and independent of various gauge-fixing parameters used at any intermediate step of computations. We add that truly physical observables, such as cross sections, are obtained from invariant squares (complex absolute moduli) of the *S*-matrix elements (amplitudes) to remove complex phases that are often too difficult to measure. This corresponds to measuring of probabilities and densities of them, but not of mere scattering or transition amplitudes.

One can prove, using standard tools of general gauge theories, that the partition function in the absence of the coupling to the external current $J^{\mu\nu}$ is completely independent of the choice of the gauge-fixing conditions $\chi_\gamma$ used to define it. The form of $\chi_\gamma$ and the matrix-differential operator $G^{\gamma\delta}$ can be arbitrarily changed and the resulting $Z_{\mathrm{grav}}$ remains the same. We only remark that for different, rather than standard and minimal, choices of these auxiliary elements needed for the definition of quantum theory, the explicit computation may be very difficult, and a much more tedious task than for minimal cases. However, the general proof holds for any local symmetries. This is a gauge-fixing independence of the true representation for the partition function $Z$ without coupling to an external current. When one adds this coupling of the form $h_{\mu\nu}J^{\mu\nu}$, then all Green functions (such as propagators or effective higher-order vertices) are gauge-dependent objects. This happens entirely due to the term $h_{\mu\nu}J^{\mu\nu}$ added to the total action $S_{\mathrm{tot}}$. At the end, we indeed compute higher *n*-point functions using the functional $Z[J^{\mu\nu}]$ with the current $J^{\mu\nu}$ turned on and by consecutive variational differentiations with respect to it (and only typically after having performed the variations, setting it to zero value). Finally, due to the mentioned equivalence theorems due to Weinberg, on the level of single elements of the *S*-matrix, we see full gauge-independence and also full gauge-fixing-independence. This last one signifies that nothing there depends on arbitrary elements $\chi_\gamma$ and $G^{\gamma\delta}$ of the FP construction related to fixation of gauges for all local symmetries present in the models under study.

## 4. Early Attempts of Quantization of Conformal Symmetry

### 4.1. A Need for Conformal Gauge-Fixing

Now, we discuss the issues related to conformal gauge-fixing. This parallels the discussion of fixing the gauge for diffeomorphism symmetry in general HD gravitational theories. This issue is at the heart of covariant and perturbative quantization of conformal gravity in any even dimension $d > 2$. We just emphasize that since local conformal symmetry is a different symmetry than diffeomorphisms, then on the level of FP quantization it will come with a different set of gauge-fixing conditions, different set of ghosts, different weighting functionals, and different kinetic operators in the ghost sectors. We will, in what follows, distinguish these new objects related to conformal symmetry from ones related to diffeomorphism symmetries by appending the subscript "c" on related quantities in the formalism of covariant FP quantization for conformal symmetry. Besides this, there is an obvious fact that the action for such a conformal gravitational theory is different than a generic action of globally scale-invariant HD gravitational theory in any even dimension $d > 2$ of spacetime. We also emphasize that here it is not enough to constrain and gauge-fix only diffeomorphism symmetry of an HD gravity, since the conformal symmetry is in the local version too, and this local conformal gauge invariance has its usual consequences

as any local symmetry has on the formalism of quantization and also on the structure of field configurations and physical representatives of fields there. The only similarity and common overlap between these two local symmetries is that they will use the same field as their gauge potential—namely, the covariant tensor of metric fluctuation field $h_{\mu\nu}(x)$, on which transformations act and which will become a proper quantum variable. This is because we do not introduce here the Weyl gauge connection field (an analogue of photon field for conformal symmetry).

First and foremost, however, we need to fix the conformal gauge in quantum conformal gravity, because the graviton kinetic operator in conformally invariant theory is degenerate, and some steps with its processing are not well-defined. This degeneracy is again the consequence of conformal symmetry present in the local version. Moreover, we have to properly extract the factor of the volume of the conformal group for the start of the successful FP quantization procedure. As it is well known in general gauge theories, the full space of configuration fields comes with a big over-baggage. This redundancy comes from the fact that gauge fields are a very redundant description of the physical system with a lot of symmetries. In actuality, in the covariant framework of quantization, the more symmetries present, the more fields need to be added, and the description is more redundant. This is in clear opposition to the case of canonical treatment of gauge theories, when more symmetric theories come with more constraints and the space of physical field configurations is effectively smaller. For example, quantum conformal gravity is a system with very particular and highly constrained relativistic dynamics. In two special dimensions, $d = 2$, the number of constraints is so big that the quantum gravitational system is practically over-constrained and it becomes a completely integrable and even solvable system of quantum (conformal) gravitational field theory.

The field space in conformal gravity, besides the redundancies related to the diffeomorphisms, has the structure of orbits related to conformal transformations. We have to find proper conformal representatives of field configurations, since in the covariant formalism, different field configurations, but related by some conformal transformations, physically describe the same configurations. Therefore, one has to remove this multiple counting of the same physical configurations in a true representation for the partition function of the quantum conformal gravitational system. In other words, one has to properly extract the volume of the conformal group, the five-dimensional part (or local part with GR scale transformations from (3)) of the full 15-element conformal group Conf in $d = 4$ spacetime dimensions. As usual in the FP procedure, this restriction of the path-integration to a physical subspace spanned only by physical representatives of gauge configurations, is achieved by a choice of the particular conformal gauge. For the consistency of the whole procedure, we must ensure that we integrate precisely once over each physical configurations, which must be different from the point of diffeomorphism symmetries, and also they cannot be conformally related.

The geometric structure of the full conformal gauge configuration space is an interesting topic. However, as often performed in physics, rather than applying very formal methods of computation, we use heuristic ones due to Faddeev and Popov. Instead of describing the structure of conformal orbits in the space of all metric fluctuation fields $h_{\mu\nu}(x)$ by picking some suitable coordinate system for this infinite-dimensional linear space, and instead of restricting the path integral only to a subspace which is physical and crosses each conformal orbit precisely once, we prefer to add a conformal gauge-fixing to the theory given by a suitable functional. We could restrict the integration to this, usually a nonlinear curved subspace of lower but still infinite dimensions and of conformal field representatives, and we could introduce geometric coordinates there. However, due to explicit and intrinsic nonlinearities present in such a natural geometric framework, we adopt a different heuristic method. We still want to integrate over the full vastly redundant configuration space, but we need then to modify the integrand of the partition function and make it different from the naive original integrand which was simply $\exp(iS_{\text{grav}})$. The modification should take into account that we want to restrict the integration to some

"physical" subspace which completely solves the gauge-fixing condition with special attention given to the fact that this submanifold may be extrinsically curved in the embedding configuration space. All these geometrically sound conditions and demands are taken fully by the addition of the FP determinant to the integrand under path integral. By properly exponentiating this FP determinant, one sees that the ensuing total action of the theory is modified and it does not consist only of $S_{\mathrm{grav}}$: there are necessary quantum additions to it with new quantum dynamical fields, new ensuing actions for them, and a gauge-fixing part for local conformal symmetry.

On the other hand, the geometric condition that each conformal orbit crosses the space of conformal representatives precisely once is typically difficult to check explicitly. One can quite easily ensure this in a small neighborhood of the first crossing point, but for further neighborhoods, this becomes more challenging. This problem was first described by Gribov in gauge theories and the resulting ambiguity brings his name, together with the possibility of having different theories with different non-perturbative vacua, so-called Gribov copies, coming from the same quantum partition function defined by the same classical theory. We will discuss the tentative solution to the Gribov ambiguity issue of local conformal symmetry in conformal gravity in one of the later publications.

As a simple choice of the conformal gauge we may choose to fix the value of the trace of the metric fluctuations

$$h = g^{\mu\nu} h_{\mu\nu} \tag{21}$$

for various quantum computations. This is one of the possible conformal gauge-fixing conditions. In principle, there could be many of such gauge conditions. We restrict our attention only to gauge conditions enforced on the metric fluctuation field $h_{\mu\nu}(x)$, since this is the gauge potential in our gravitational context available at our disposal. Moreover, we focus on linear gauge-fixing conditions which do not contain higher than the first power of the fluctuation $h_{\mu\nu}(x)$. This restricts quite a lot class of possible conformal gauges. The trace of the metric is a scalar from the point of view of flat spacetime Lorentz symmetry, hence this condition is quite simple.

It is important to note the following obvious thing: the gauge-fixing condition has to be non-invariant under transformations of symmetries that it is supposed to fix. By using a conformal invariant constructed out of the metric fluctuation field $h_{\mu\nu}$, one would not sufficiently specify any conformal gauge since arbitrary conformal transformations would still keep this invariant untouched, hence there would still be a freedom in performing them. One has to check that indeed the trace of the metric fluctuations $h$, contrary to the expressions as built in footnote 4 , transforms non-trivially under conformal transformations, and hence fixing it to some value constrains the freedom of conformal transformations to the level that none of them can be additionally performed anymore. This is detailed in the section where there is a discussion of the solution to the conformal Gribov ambiguity problem.

In general, we can form a family of gauges by requiring that

$$\chi_c = \chi_c \left[ h_{\mu\nu} \right] = h - \ell_c = 0 \tag{22}$$

for arbitrary (also spacetime-dependent) (non-dynamical) field $\ell_c = \ell_c(x)$. The choice in (22) we will call the simple conformal gauge. The choice we made for the scalar was not a vector, as a conformal gauge condition is dictated by the fact that the parameter $\Omega = \Omega(x)$ of conformal transformation is also a scalar, so it does not carry any spacetime or Lorentz index. This is an important property of the conformal transformations, in distinction to diffeomorphisms when such a choice must be made for a vector $\chi_\mu$. One can see that the vector (or, in general, any tensor of odd rank) will not work well with the FP quantization procedure since then, for local conformal symmetry, the FP operator would be with odd number of spacetime indices (two from the infinitesimal generator in (14) and three from the variational derivative of a hypothetical vector conformal gauge-fixing condition $\chi_{c,\mu}$ with respect to the metric field which is here a charged potential field for conformal symmetry;

at the end we contract two pairs of indices, but the parity remains unchanged). Of course, then it is impossible to take a determinant of such an operator and then naively the whole FP procedure fails[7]. One must also admit that the scalar conformal gauge-fixing condition $\chi_c$ is much simpler than the next theoretically possible tensorial one $\chi_{c,\mu\nu}$. For simplicity, here, we will restrict our considerations only to this first choice.

The conditions in family (22) are linear admissible gauges, but they do not contain any derivatives acting on the potential field $h_{\mu\nu}$. Of course, this choice is also dictated by simplicity, however one can easily complicate the simple choice from (22) by inserting in the middle some differential operator constructed with the background metric and being covariant with respect to it. For example, we can use GR-box operator $\Box = g^{\mu\nu}\nabla_\mu\nabla_\nu$ or simply two derivatives $\nabla_\mu\nabla_\nu$ acting on the field of metric fluctuations $h_{\mu\nu}$. By the arguments stipulated above, it is obvious that we must use an even number of derivatives. These are more complicated gauges; however, they may have some advantages over simple conformal gauges. We will also show later that the physical results properly do not depend on this choice of the conformal gauge-fixing conditions.

*4.2. Hidden Local Conformal Symmetry and Veltman-like Discontinuities*

According to the general philosophy, one must be very careful with all local symmetries present in a model that is being prepared for quantization, both in the visible (manifest) form and also in invisible forms. For example, if one superficially looks at the theory

$$S = \int d^4x \sqrt{|g|} \left( R_{\mu\nu}^2 - \frac{1}{3}R^2 \right), \tag{23}$$

then one sees only diffeomorphic symmetries of general relativity and a theory with HD gravitational action. One does not recognize that this action is actually conformally invariant since the usage of Gauss–Bonnet identity in $d = 4$ dimensions valid under spacetime volume integral reduces manifestly conformally-invariant-looking action with the square of the Weyl tensor $C^2$ to this form above in (23). However, the conformal symmetry is still present there but in a hidden form. It is still there in a local version and it should be always properly treated and gauge-fixed for the quantization purposes. One sees the problems with a naive interpretation of the action in (23), if one attempts to compute a propagator for gravitational fluctuations in such a theory. The standard tricks with minimal gauge-fixing for (only) diffeomorphisms do not work here since after them the kinetic operator for small fluctuations is still degenerate. One may rediscover in this way that the theory is with enhancement of symmetries and that these symmetries need a special treatment—basically, the hidden conformal symmetry should be revealed. In conclusion, no matter what our knowledge is of the symmetries of the theory, we always have to be prepared that for every not-manifest symmetry that we find enhanced and hidden, we must perform a proper separate FP quantization procedure (a special treatment of local gauge symmetries) if we want to move to a quantum level with such a theory.

All gauge symmetries of a given model should be treated with care, gauge-fixed, and their violations by inclusion of explicit gauge-fixing conditions should be controlled in order not to produce any unwanted anomaly of gauge symmetries on the quantum level. The latter ones are signaling that something very bad is happening with the quantum version of the theory and that the theory is sick and sustains itself only on the classical level and does not survive the quantization procedure. This is our general philosophy in this paper to treat all local gauge symmetries with their due respect, and they should not be neglected (or overlooked) for the quantization process, even if one knows *a posteriori* that these local symmetries cease to exist on the quantum level (for example, due to anomalies or explicit breaking), but this is caused by some other physical circumstances related to other physical mechanisms or phenomena. Basically, we do not want the quantization act by itself to produce some fatal violations of local gauge symmetries that cannot be recovered on the quantum level. We hope that all local symmetries are here with adequate respect and that their breaking is performed in a fully controllable manner. This is similar

to what is performed, for example, with the covariant treatment of gauge symmetries in standard YM theories, where one has full control (due to the introduction of gauge-fixing parameters). One introduces gauge-fixing conditions, properly parametrized, and as a result, the gluon propagator in some gauges is able to be derived, allowing for computation of more complicated Green functions at loop levels and off-shell. However, at the end, one recovers on-shell quantum physical results as gauge-invariant, gauge-independent, and gauge-fixing-independent. We want to apply the same attitude, following from a successful FP quantization when applied to YM theories, here to other symmetries in the local form, in particular to conformal symmetry present in the gravitational context.

The action of conformal gravity in $d = 4$ spacetime dimensions is typically taken as $S = \int d^4x \sqrt{|g|} C^2$, since in this form conformal invariance is manifest. One can perform integration by parts and neglect (or add—depending on the point of view) some boundary terms. They are important for the canonical approach to quantization of gravitational gauge theories. By neglecting the Gauss–Bonnet (GB) density term (see also (31)), which integrates to the boundary term, one can rewrite the original action in the form

$$S = 2 \int d^4x \sqrt{|g|} \left( R_{\mu\nu}^2 - \frac{1}{3} R^2 \right). \tag{24}$$

Its properties under conformal transformations are the same (up to the boundary terms, so one neglects the derivatives of the parameter $\Omega(x)$ on the boundary). The action as written above has simplicity and one notices the fact that now one removes the terms with Riemann tensor. Since it is a completely equivalent action on the classical level, then one can study its exact solutions. For example, in gravitational vacuum situations, one finds immediately that all Ricci-flat solutions are also exact vacuum solutions of this higher-derivative theory [13].

Besides this nice feature of conformal gravity in $d = 4$, the reason for using the action of conformal gravity in the form (24) is also of the more theoretical nature. One keeps neglecting the GB terms presently hidden in the original action with Weyl tensors. The reason for doing this is evident, when the canonical charges [86] subject to appropriate boundary conditions are constructed in this theory. It is worthwhile to mention that the surface terms as present implicitly in (24) render the conformal gravity action (24) to have well-defined functional derivatives, under some suitable boundary conditions (such as, for example, on AdS asymptotics [87,88]). Not surprisingly, the resulting total Hamiltonian derived from the form of the action in (24) is already an improved gauge generator under these boundary conditions, so then it does not need an improvement by surface integrals. Additionally, due to the specific form of one of the secondary constraints, the extended Hamiltonian arising from (24) also has well-defined functional derivatives under these boundary conditions. Therefore, the inclusion of boundary terms and usage of the action as in (24) is more favored theoretically, since asymptotic charges of conformal gravity are then clearly extracted. We remark that the Hamiltonian here is the generator of all local gauge transformations present in the system (diffeomorphisms and local conformal (Weyl) rescalings).

According to what we said before, we would like to have a quantum theory resulting from the covariant quantization procedure such that it treats well (as much as possible) the local conformal symmetry present in some gravitational models. The gauge symmetry has to be fixed, but at the same time its violation has to be fully controlled by gauge-fixing parameters, which later drop out from physical observables. This is the essence of the covariant quantization act, but the quantization procedure is not the end of the story on the quantum level. It is well known that one also has to regularize the theory on the quantum level due to UV-divergences arising due to perturbative loop diagrams. Such regularization procedures also have to satisfy some stringent conditions. In particular, they have to preserve as much symmetry of the original classical theory as possible. Therefore, they have to be selected from a set of covariant regularization procedures. They must not touch the gauge symmetries of the model; for example, they cannot be based on sharp

ultraviolet momentum cutoff. The preferred regularization for ordinary gauge theories is dimensional regularization or covariant $\zeta$-function regularization.

However, in the setup of a gravitational theory also enjoying conformal symmetry, one cannot play with the dimensionality of regularized spacetime, since the dimension is a very delicate feature there and conformal symmetry depending on the form of a gravitational action is realized only in a fixed number of dimensions. They cannot be even perturbed by a small $\varepsilon > 0$. Other covariant regularization schemes for UV-divergences have to be employed in quantum conformal gravity to regularize these divergences in a fully conformally invariant manner. Moreover, one requires that the counterterms should be written in a conformally covariant way, so the divergent part of the effective action is also a conformally invariant functional in a fixed number of dimensions. Then, in such a setup of quantum conformal gravity, one could say naively that the conformal symmetry is preserved by the UV-regularization procedure and the effective action in quantum theory also seems to look as with full conformal symmetry realized[8]. In conclusion, both the quantization procedure and the regularization of ultraviolet physics in quantum field theory approaches to conformal gravity have to treat the delicate local conformal symmetry gently.

Let us take, in particular, the example of conformal gravitational theory in $d = 4$ spacetime dimensions. The action for this theory was introduced in Section 2 in (16). One may think that this is a very special case of a generic four-derivative theory which is scale-invariant precisely in $d = 4$ dimensions. The action of such a HD gravitational theory can be conveniently written as

$$S_{\mathrm{HD}} = \int d^4 x \sqrt{|g|} \left( \alpha_{R^2} R^2 + \alpha_{C^2} C^2 \right), \tag{25}$$

where we have deliberately chosen the basis with curvature invariants $R^2$ and $C^2$, which make explicitly manifest the conformal covariance of the part with the $C^2$ term there[9]. One would think (naively) that to specify to a conformal gravitational theory in $d = 4$, one has to simply take a limit $\alpha_{R^2} \to 0$ from the generic HD theory in (25).

One notices that for generic HD gravitational theories such as in (25), there are no problems or subtleties with the FP quantization procedure of the existing diffeomorphism local symmetry, which can successfully be brought to the very end here, and, for example, the propagator around flat spacetime of graviton or covariant functional traces of the Hessian operator understood around general background can be computed. In general, the method of FP quantization of higher-derivative gravitational theories (with local diffeomorphism symmetries) works for the cases when both $\alpha_{R^2} \neq 0$ and $\alpha_{C^2} \neq 0$. One may think that the results of the same procedure in the case when the limit $\alpha_{R^2} \to 0$ is taken will be smooth and continuous ones.

However, this is not the case and one sees some strong discontinuities in the details of this procedure as well as in the final results obtained within the QFT framework. One can see this discontinuity already in the perturbative spectrum of the two theories (with $\alpha_{R^2} \neq 0$ and with $\alpha_{R^2} = 0$) when analyzed around flat spacetime. As it is well known in general higher-derivative gravitational theories with four derivatives, one typically has eight degrees of freedom, where also the Einstein–Hilbert term $\kappa R$ with the Newton's constant is included into the action in (25). These results are derived either from the canonical formalism of treatment of theories with gauge symmetries and performed by counting constraints [89], or by taking a covariant perturbative approach and looking at the spectrum of modes which solve linearized EOM of the theory. A similar procedure can be also applied there to Weyl conformal gravity. However, there are more subtleties due to new constraints due to local conformal symmetry in the canonical framework and due to the masslessness character of all fluctuations considered in the perturbation theory (some poles of the propagator are double and need an additional resolution). But, using some tricks with well-designed limiting procedures [90,91], one obtains from both the two methods the consistent result, namely six degrees of freedom in conformal gravity in $d = 4$. This last number of degrees of freedom in conformal gravity is unambiguous. As one

can see, this result is clearly discontinuous from the other one (eight degrees of freedom) obtained for any non-zero value of the coupling $\alpha_{R^2}$ and for $G_N \neq 0$. Hence, the limit $\alpha_{R^2} \to 0$ will also not change anything and we inevitably find a discontinuity.

### 4.3. Counting of Degrees of Freedom

The full story of counting active degrees of freedom in two theories (generic HD gravity in (25) and Weyl conformal gravity in $d = 4$) is, however, more complicated and more twisted. For scale-invariant HD theory, one should never include the $\kappa$ coupling related to the Newton's constant multiplying the Ricci scalar $R$ term in the Einstein-Hilbert action. This is obvious from the fact that this dimensionful parameter gives rise to masses of perturbative modes, which are, however, strictly forbidden in theory without scales. On the other side, the $\kappa$ coupling is technically needed to resolve the double poles of the graviton propagator, otherwise it is problematic both in generic HD theory and in conformal Weyl gravity too. The standard particle interpretation is only available for modes whose propagation after simple fraction decomposition is described by simple poles in the $k^2$ variable. To resolve the multiple poles, one tries to modify the theory and introduce a spurious massive parameter $M$ (breaking conformal and even scale invariance). At the end of the computation, one should, as the last limit, take $M \to 0$ hoping that for physical observables this limit will be unambiguous and possible to define. In this way, the problem with multiple poles can be resolved and physical quantities can be determined.

If instead one naively counts possible number of degrees of freedom in massless theory (HD theory in (25) with $G_N = 0$), then one may have at most six degrees of freedom ($2 \times 2 = 4$ from two massless poles of the spin-2 part of the propagator and $2 \times 1 = 2$ from two massless poles of the spin-0 part). This final number is in clear disagreement with the number obtained using other methods, also when $\kappa = 0$ (such as by counting dynamical constraints or computing one-loop functional determinants in scale-invariant HD theory). However, as one of the advantages of the previous method, we notice that we do not anymore play with the mass to resolve the double poles and we do not inadvertently violate the scale invariance by intrusion of mass to the definition of the theory and perturbative states around flat spacetime.

Still, assuming that in generic scale-invariant HD theory in (25) we have eight perturbative degrees of freedom, it is difficult in a standard way to assign these degrees of freedom to particular spin states. This is because we do not have mass in the theory, so we cannot have massive spin-2 excitations either, and this forbids us from having $2 + 5 = 7$ degrees of freedom in the spin-2 sector. In the massless spin-2 sector with two modes, we can have at most four degrees of freedom, while in the spin-0 sector we can have at most only two degrees of freedom when we have two poles. This does not sum up to eight. Instead, the maximal number of degrees of freedom we can obtain in this way is six. This different way of counting shows that there may not be such strong discontinuity when one takes the limit $\alpha_{R^2} \to 0$ from generic massless HD gravity to a special conformal Weyl gravity in $d = 4$ spacetime dimensions. In actuality, one may think that since in this case of counting, the number of active degrees of freedom is continuous $6 \to 6$, then the limit $\alpha_{R^2} \to 0$ is not so dangerous and signifies the mere different accounting or reshuffling of degrees of freedom. One could be tempted to say that in generic scaleless HD gravity in $d = 4$, we have six degrees of freedom distributed as two spin-2 massless particles, each with two polarizations, and two spin-0 massless particles. When the transition to conformal gravity happens, the two scalar degrees of freedom (from spin-0) merge into one massless spin-1 vector particle with two helicities. It is clear from this counting that the number of degrees of freedom is conserved in this limiting procedure. They just obtain different specification and distribution in conformal gravity into slightly different irreducible representations of Lorentz group describing massless particle perturbative excitations around flat spacetime. The emerging massless vector spin-1 gauge particle is reminiscent of the Weyl gauge photon, but here, its role in the gravitational context is completely different.

Due to additional constraints and symmetries of the conformal gravitational theory, one also makes visible the discontinuity in the structure of the two theories. New symmetries are present in the special theory with $\alpha_{R^2} = 0$. The discontinuity in the number of degrees of freedom signals that something is very special with the theory characterized by the absence of the $R^2$ term in (25). The symmetries are enhanced in this case, the limit is not continuous, and this is not similar to any other generic HD gravitational theory. This point in the theory space (actually a line parametrized by any value of the coupling $\alpha_{C^2} > 0$) is very special and it is isolated from its neighborhood; hence, any limiting procedures do not give any continuous results here. Moreover, one can view that the conformal gravity is a critical or extremal point in the theory space, not reachable by any sequence of generic HD theories asymptoting to it. We already know the reason for such a behavior, since the symmetries are enhanced and the conformal symmetry of rescaling transformations appears there in the local form.

As is known, the counting of degrees of freedom can be also performed in canonical (Hamiltonian) formalism for gauge field theories. This formalism [86] is also very useful as a starting point for canonical quantization of theories with degeneracy of the Lagrangian, so when not all of the generalized velocities can be re-expressed via generalized momenta, this generically happens due to the presence of symmetries (in local form). The freedom related to gauge symmetries typically indicates that there is more than one set of canonical variables that corresponds to the same physical state. This choice can be realized by adding some gauge-fixing conditions. General gauge theory can be viewed as a mechanical theory with constraints, since not all canonical momenta are non-zero during the spacetime evolution of fields. For example, these constraints are analogues of the Gauss laws for the electric part of the field strength tensors. The constraint is of the first class if its Poisson bracket with every other constraints vanishes on the constraint surface, but not necessarily in the full phase space. Then one could say that they vanish weakly. If Poisson bracket of a constraint with at least one other constraint does not vanish even weakly, then they are of the second or higher class. Typically in gauge theories, first-class constraints play the role of generators of local gauge symmetries. The same interpretation cannot be applied to the second-class constraints since they do not preserve constraints even on the constraint manifold. They are non-physical and they have to be treated differently to generators of gauge symmetries. The second class constraints can be set strongly to zero (so valid in the whole phase space and not only on the constraint surface) either before or after the evaluation of a Dirac bracket, which here substitutes for the Poisson bracket for the considerations of the extended Hamiltonian function. They just become identities which must be satisfied during the field evolution, expressing some canonical variables in terms of others.

In the case of conformal gravity, setting second-class constraints strongly equal to zero can actually be used to eliminate some of the canonical dynamical variables of the formalism. In this way, one can completely select the gauge by eliminating all first-class constraints and ending up only with second-class constraints. Following this approach, one arrives at the following counting of physical degrees of freedom (understood as field-theoretical degrees of freedom, contrary to mechanical ones) $N_{\mathrm{dof}}$ in generic gauge field theory:

$$N_{\mathrm{dof}} = N_{\mathrm{can}} = N_{\mathrm{tot,can}} - N_1 - N_2 - N_{\mathrm{gfix}} = N_{\mathrm{tot,can}} - 2N_1 - N_2, \tag{26}$$

where by $N_{\mathrm{can}}$ we denote the number of independent canonical variables in the formalism. Moreover, $N_{\mathrm{tot,can}}$ stands for total number of canonical variables (possibly mutually dependent) and $N_1$ and $N_2$ are for the number of first- and second-class constraints, respectively. Finally, by $N_{\mathrm{gfix}}$ we mean the number of gauge-fixing conditions added to unambiguously pinpoint the dynamics of the system in question. This counting in the case of Weyl gravity is as follows [92]: the physical degrees of freedom are $N_{\mathrm{dof}} = 12 - 3 - 0 - 3 = 12 - 6 = 6$ (according to (26)). The interpretation of these resulting six physical degrees is as follows: two of these describe a massless graviton, such as in the case of two-derivative general relativity, and the remaining four can be used for a description of a "partially massless"

graviton, as in [93,94]. Following appendix C.3 of [92], one can see that due to the higher-derivative nature of the conformal gravity, two additional primary constraints are actually of the second class; however, one can set them strongly to zero, and they are effectively eliminated from the Lagrangian of the conformal gravity. In other words, these second-class constraints can be solved everywhere in the phase space. This also implies that instead of the extended Dirac bracket, one can use, for the extended Hamiltonian, simple Poisson bracket here without too much complication.

In these above two paragraphs, the conformal gravity theory was formulated in the Hamiltonian formalism. The original work can be found in [87]. Earlier, a Hamiltonian formulation of generic higher-derivative theories in $d = 4$ spacetime dimensions, including conformal gravity, was considered in [50,89].

The general fact of discontinuity is an emanation of a quite ubiquitous phenomena which was first discovered by Veltman and van Dam in 1970 [95]. When some parameters of the theory reach critical values, then the limits are discontinuous and the resulting theory can be very special. Of course, originally Veltman discovered these discontinuities in massive vector theories when taking the mass parameter (which from the nature of things is dimensionful) to zero, which changes the character of symmetries present in the theory, and local gauge symmetry arises. This new symmetry and new dynamical situation is not possible to predict, being always within massive vector models. Of course, there we also have the discontinuities in the number of degrees of freedom, namely, three in the massive case and two in the massless situation with enhanced gauge symmetries. One may think that it is the gauge symmetry which kills the propagation of one longitudinal mode of the massive vector gauge boson. In the case when $m^2 = 0$, the dynamics are different and there is an enhancement of symmetries as well. The perturbative spectrum is different and also the quantum correlation functions differ. One also understands well the reason behind this case, because then local gauge symmetry is born. However, one could view sending the mass parameter to zero as a very crude procedure. Similar considerations in the gravitational setup were also given to transition: massive gravity into massless gravity, first by Veltman, van Dam, and Zakharov. The discontinuities in that case bear their names. The reasoning is exactly the same as in the case of massive and massless gauge theories analyzed already by Veltman. In the gravitational context, the local enhanced symmetry of the massless gravitation here kills three "longitudinal" states of the previous massive spin-2 graviton, morphing it into a massless spin-2 graviton with only two polarizations.

We emphasize that in the case of various limits of HD theories, and in particular of the limit $\alpha_{R^2} \to 0$, the situation is more delicate since the coupling parameter here is dimensionless. Hence, its limit is not as rough as for a dimensionful parameter. One actually may expect that these kind of limits are better behaved and are continuous. This is still not the case for the transition to conformal gravity from generic HD gravitational theory, since the number of degrees of freedom changes abruptly, that is, this symmetry, when present, modifies the perturbative spectrum of active degrees of freedom. This could be viewed as a more subtle form of the Veltman discontinuity; however, it is still a discontinuity. We see that, in principle, it is possible to distinguish two types of Veltman-like discontinuities. When the transition from scale-invariant HD gravity to conformal gravity occurs, we see a softened version of discontinuity. The more strong is when we modify the HD gravity from (25) by adding the dimensionful parameter, as in with the Einstein–Hilbert term. The situation with both these limits is not so clear because one time they are treated as technical tricks, but the other time they change the counting of degrees of freedom and interfere with symmetries of the models in question. One still sees that the limit to conformal gravity $\alpha_{R^2} \to 0$ is very special and delicate and should be approached with great care, as the last one in the computation procedure, where the order of different limiting procedures matters.

This unpleasant situation with the problems of definition of spectrum of perturbative modes around flat spacetime happens also for conformal gravities in higher dimensions of spacetime, and here, the case of the $C^2$ gravity from $d = 4$ is not an exception. In general,

even in dimensions $d$ of spacetime, the Lagrangian of conformal gravity may be found in the schematic form

$$\alpha_{C^2} C \square^{(d-4)/2} C + O\left(C^3\right), \tag{27}$$

while the Lagrangian of scale-invariant respective HD and quadratic (in curvatures) gravity there has the similar form

$$\alpha_{C^2} C \square^{(d-4)/2} C + \alpha_{R^2} R \square^{(d-4)/2} R. \tag{28}$$

However, the same problems with limits (such as $\alpha_{R^2} \to 0$), discontinuities, disagreements in number of active perturbative degrees of freedom, and resolutions of poles in the propagators are met in this higher-dimensional setting with only modifications of different counting of degrees of freedom in different dimensions.

All the arguments above again and again confirm that the theories with local conformal symmetry have to be treated specially and no naive limiting procedures can reproduce their results. Since the symmetry is the mathematical embodiment of the beauty of nature, this simply means that theories with local conformal symmetries are discontinuous from any other scale-invariant HD theories in (25). This fact of discontinuity is true, but it makes the conformal gravitational theories even more beautiful.

Let us repeat that the conformal gravity is special and is not obtainable by a continuous limit from a general Stelle theory, when $\alpha_{R^2} \to 0$. There is an enhancement of symmetries in this critical point, where the $R^2$ term is not present for Stelle four-derivative gravitational theory. This is an enhancement to full local conformal symmetry on the classical level, and on the first loop quantum level (up to the issue of conformal anomaly). This theory with enhanced symmetry should be treated specially and differently to generic HD theories. First, the FP method for fixing conformal gauge symmetry (with proper conformal gauge-fixing and additional conformal sector of ghosts fields) should be used in conformal gravitational models. In such a way, the quantization procedure is performed as conformally covariant as possible and with due care given to local conformal symmetry. Finally, conformally invariant regularization procedure should be employed to define quantum theory, especially to take care of perturbative UV-divergences. If all these requirements are satisfied then we can announce an achievement of a construction of quantum theory with local conformal symmetry conserved in field theory models. The physical predictions in such quantum theories preserve the underlying local conformal symmetry as much as is possible and dictated by the quantum nature of the model. The last is shaped and controlled exclusively by nature and completely independently of what regards the formalism used by theoreticians to describe the nature and to perform the artificial acts of quantizations or regularizations within given models.

## 5. Fradkin–Tseytlin Approach to Pure Conformal Gravity in $d = 4$

In the context of a limiting procedure from generic HD gravitational theories in (25), one must notice the paper by Fradkin and Tseytlin from 1982 [96]. They worked in the four-dimensional setup and analyzed perturbative one-loop UV-divergences of some HD theories generically described by the action in (25). Their computation is fully well-defined for the case when $\alpha_{R^2} \neq 0$ and $\alpha_{C^2} \neq 0$. To perform the covariant quantization process they use the general FP method for diffeomorphisms, as elucidated at length before. Here, this method was applied to higher-derivative gravitational theories. For extracting UV-divergences, they use general diffeomorphism, preserving regularization of QFT of the HD gravitational model. Their computation gives correct results in the generic case when $\alpha_{R^2} \neq 0$ and $\alpha_{C^2} \neq 0$. However, as one of the additional developments, they wanted also to consider the conformal Weyl gravity as a special case of HD theories and also as the limit of HD theories, when $\alpha_{R^2} \to 0$. Of course, the authors were aware of the discontinuity present in such a limit. However, they also knew that the model, in the case when precisely $\alpha_{R^2} = 0$, has enhanced symmetries, and this new symmetry, on the classical level before quantization, was the local conformal symmetry of conformal Weyl gravity.

*5.1. The Limit $\alpha_{R^2} \to 0$*

The strategy that authors of [96,97] followed was the following. Knowing that the conformal symmetry was present at the classical level of the theory with $\alpha_{R^2} = 0$, they also expected it to be present in some unbroken form on the quantum level, here analyzed to the level of one-loop accuracy. This assumption allowed them, as we will explain later, to use the power of conformal symmetry at the end of their computation for clearing the situation with counterterms of the UV-divergent one-loop effective action. As a matter of fact, these counterterms in $d = 4$ are all globally scale-invariant for generic HD gravitational theories. They are, namely, of three possible forms here; $R^2$, $C^2$, and $E = $ GB, where the last one is a Gauss–Bonnet scalar term.

As analyzed before, the Weyl squared term, given explicitly by

$$C^2 = C_{\mu\nu\rho\sigma}C^{\mu\nu\rho\sigma} = R_{\mu\nu\rho\sigma}R^{\mu\nu\rho\sigma} - 2R_{\mu\nu}R^{\mu\nu} + \frac{1}{3}R^2, \tag{29}$$

when properly densitized to the form $\sqrt{|g|}C^2$, is a conformal invariant. That is, we have for infinitesimal conformal transformations (denoted below by $\delta_c$), which only matter here, that

$$\delta_c\left(\sqrt{|g|}C^2\right) = 0. \tag{30}$$

On the other hand, the last invariant is topological and is known as the celebrated Gauss–Bonnet scalar (or as an Euler density $\sqrt{|g|}E$ in a densitized form). Its expansion, in other terms quadratic in standard gravitational curvatures, reads,

$$\text{GB} = E = R_{\mu\nu\rho\sigma}R^{\mu\nu\rho\sigma} - 4R_{\mu\nu}R^{\mu\nu} + R^2. \tag{31}$$

Since it is a topological term (depending only on the boundary of spacetime), then any of its bulk variation carried out in the volume of spacetime (in particular the conformal infinitesimal one $\delta_c$) vanishes, that is, we have

$$\delta\left(\int d^4x \sqrt{|g|}\text{GB}\right) = 0 \tag{32}$$

as an identity of variational four-dimensional calculus.

Finally, the last invariant with respect to diffeomorphisms (scalar) out of the group of three local invariants in curvature in quantum gravity, that is, quadratic in the curvature, is simply the square of the Ricci scalar $R^2$. Here, one notices an interesting thing pertaining to conformal transformations of this invariant which is quadratic in gravitational curvatures (similar to Riemann tensor, Ricci tensor, and Ricci scalar). This term is globally conformally invariant in $d = 4$, i.e., it is scale-invariant because its energy dimension as the operator is $\dim(R^2) = 4$. The coupling coefficient in front of it in the action in (25), this $\alpha_{R^2}$, is of course a dimensionless parameter in $d = 4$ spacetime dimensions. This fact is common also for all the other invariants considered above (in (29) and (31)), and this happens precisely because all these invariants contain four derivatives on the metric tensor and this number coincides here with the dimension of spacetime. Hence, the $R^2$ term, $C^2$ term in (29), and the GB term in (31) are properly scale-invariant terms when densitized on the level of action functional. As we have seen above for the last two, we have upgrade of these symmetry invariance properties to the full invariance under local conformal symmetry understood in the GR framework.

For the first term $R^2$, however, the situation is different and its symmetry invariance obtains some promotion but not to the full conformal invariance. Namely, the $R^2$ term is invariant in $d = 4$ dimensions under so-called restricted conformal transformations, that is, the transformations from (9), satisfying the additional condition that

$$\Box\Omega(x) = 0, \tag{33}$$

which is a kind of GR-covariant wave equation (GR-covariant d'Alembert equation) for the parameter of the conformal transformations $\Omega(x)$. This means that this parameter $\Omega = \Omega(x)$ can be spacetime-dependent here (already this is more than a case of invariance under global transformations when we have that $\Omega = \text{const}$), but this dependence cannot be arbitrary here. It must satisfy an analogue of the wave equation, which in Euclidean signature would require the $\Omega$ field again to be constant. Instead, in the Minkowskian signature, this wave equation for $\Omega(x)$ does have some non-trivial spacetime-dependent solutions. This signifies that the condition of restricted conformal invariance is more general than global scale invariance in GR with constant parameter $\Omega = \text{const}$, but still it is less than a general invariance under fully local conformal symmetry with completely arbitrary spacetime dependence in $\Omega(x)$.

One can easily see this fact by recalling the following properties of conformal transformations of the Ricci scalar. For full generality, we write them in arbitrary dimensions $d$. We have explicitly, that under local conformal transformations,

$$R \to R' = \Omega^{-2}\left(R - 2(d-1)\Box \ln\Omega - (d-2)(d-1)\nabla_\mu \ln\Omega \nabla^\mu \ln\Omega\right) \qquad (34)$$

and then using the Leibniz property of the box operator: $\Box f^2 = 2f\Box f + 2\nabla^\mu f \nabla_\mu f$, and the fact that $\nabla_\mu \ln\Omega = \Omega^{-1}\nabla_\mu\Omega$, we rewrite above as

$$R \to R' = \Omega^{-2}\left(R - 2(d-1)\Omega^{-1}\Box\Omega - (d-4)(d-1)\Omega^{-2}\nabla_\mu\Omega\nabla^\mu\Omega\right). \qquad (35)$$

Now, exactly in $d = 4$ (and only in $d = 4$, neglecting the degenerate case of $d = 1$ where $R = 0$ automatically), we see that the last term in the bracket in the above formula disappears, and we are left with the following simple local conformal transformation law in $d = 4$, that

$$R \to R' = \Omega^{-2}\left(R - 6\Omega^{-1}\Box\Omega\right), \qquad (36)$$

which indeed shows that for restricted form of conformal transformations satisfying that $\Box\Omega = 0$, we obtain that the Ricci scalar transforms fully covariantly under such local conformal rescalings. In actuality, one sees that this is the special property of $d = 4$ spacetime dimensions, since in other dimensions even restricted conformal transformations do not make Ricci scalar transform conformally covariantly because of the second term in (35) multiplied by the factor $(d-4)$ and by a mixed term $(\nabla\Omega)^2$. From this, one easily concludes the transformation law

$$\sqrt{|g|}R^2 \to \left(\sqrt{|g|}R^2\right)' = \sqrt{|g|}\left(R - 6\Omega^{-1}\Box\Omega\right)^2, \qquad (37)$$

which shows that, indeed, for restricted conformal transformations, the densitized term $\sqrt{|g|}R^2$ is (partial) conformal invariant.

### 5.2. Final Conformal Transformation

Equipped with the above knowledge, we can discuss the trick and the method used by Fradkin and Tseytlin to obtain, a bit heuristically, the UV-divergences of conformal Weyl gravity in $d = 4$ dimensions. The computation for the $\alpha_{R^2} \neq 0$ case showed that, generally, the $R^2$ term is present in UV-divergences, and this term is not conformally invariant as a counterterm, when we mean full arbitrary conformal transformations. Therefore, its presence violates the assumed conformal symmetry at the one-loop level. It would be desirable for consistency of the theory to make it vanish. We explain below what was performed to achieve this.

In an HD theory from (25), the one-loop UV-divergent part of the effective action when the limit $\alpha_{R^2} \to 0$ is taken explicitly reads,

$$\Gamma_{\text{div}}^{1-\text{loop}} = -\frac{1}{(d-4)(4\pi)^2}\int d^4x\sqrt{|g|}\left(\frac{133}{20}C^2 + \frac{5}{36}R^2 - \frac{196}{45}\text{GB}\right), \qquad (38)$$

where we effectively used the dimensional regularization scheme (DIMREG), and whole divergences are written as singular expressions in the deviation from $d = 4$ spacetime dimensions. As obvious from above, the $R^2$ term in this naive procedure is generated at the one-loop level; therefore, conformal symmetry seems to be broken by quantum corrections.

Assuming, in contrary and *a posteriori*, that the local conformal symmetry was present also on the level of the first quantum loop, the authors decided to, after the computation, take the limit $\alpha_{R^2} \to 0$ in a naive way and eventually performed a compensating additional conformal transformation, as in (9). They simply wanted to exploit the freedom of making arbitrary post-computation conformal transformations on the results of the non-conformally covariant calculation. The parameter of such a conformal transformation $\Omega(x)$ could be chosen at one's will. The authors chose such $\Omega(x)$ that $R' = 0$ in (36), that is, the result of final conformal transformation completely nullifies the presence of the square of the Ricci scalar $R^2$ counterterm. Exactly this term before was spoiling the conformal invariance of the full set of counterterms. In this way, and somehow by hand manipulations, the non-fully conformal counterterm $R^2$ is completely cancelled out, so the problem with the desired and originally-not-present conformal invariance of the UV-divergent action in conformal gravity at one-loop is solved.

For consistency, the following question then arose: Is it possible to find solutions for the spacetime dependence of the $\Omega(x)$ parameter of conformal transformations in (9), such that indeed we find that in the result $R' = 0$? The solution of the related equation in $d = 4$,

$$R - 6\Omega^{-1}\Box\Omega = 0 \tag{39}$$

or to an equivalent equation,

$$\left(\Box - \frac{R}{6}\right)\Omega = 0 \tag{40}$$

exists and it is given for $\Omega$ by

$$\Omega = \Omega(x) = 1 + \left(\Box - \frac{R}{6}\right)^{-1}\frac{R}{6}. \tag{41}$$

In the last formula, we performed some formal manipulations with treatment of the operator $\Box - \frac{1}{6}R$ as completely algebraic, although of course it is a differential operator, and taking its inverse requires some additional care. One can convince oneself that the solution for $\Omega$ in (41) satisfies the equation (40), when one notices that $(\Box - \frac{1}{6}R)$ when acting on 1 gives $-\frac{1}{6}R$ as the result of the operatorial linearity, and moreover $\left(\Box - \frac{1}{6}R\right)^{-1}$ is treated here as the operatorial inverse of the operator in the last parentheses. In particular, to define the inverse of such shifted GR-box covariant operator by possibly spacetime-dependent Ricci scalar, understood here as an arbitrary scalar field $R(x)$ on a curved manifold, one must resort to operating with Green functions of the operator since only then can we sensibly speak about the inverse on the general curved backgrounds. Moreover, the last formula in (41) is valid only when the metric $g_{\mu\nu}$, whose Ricci scalar $R = R(g_{\mu\nu})$ appears there, is asymptotically flat.

Therefore, one concludes that the solution for $\Omega$ in $d = 4$ always exists for any form of the spacetime-dependent Ricci scalar $R = R(x)$. However, the price is the resulting strong non-locality in the expression for the $\Omega = \Omega(x)$ in (41), which is the parameter of the conformal transformation that has to be performed at the end to bring back the conformal invariance to the theory on the level of one loop. This is how the authors of [96] recover, at the end, conformal invariance in the model under studies on the quantum level. It is well understood that this procedure looks very *ad hoc*. This is so because the formalism they used was suitable only for dealing with diffeomorphisms as local symmetries, and the regularization that they used was only preserving this last symmetry. The care was not exerted to properly treat the local conformal symmetry and to gauge-fix it and to add appropriate FP determinant and FP ghosts for conformal symmetry. At the end, the

authors performed quite arbitrarily-looking conformal transformations somehow to cure their results obtained in the limit $\alpha_{R2} \rightarrow 0$. For this, they had to know *a priori*, but not *a posteriori*, that conformal symmetry survives on the quantum level. Somehow, they knew it and understood why the conformal symmetry is still present on the level of one-loop counterterms and they worked out an ingenious solution and procedure to improve on their non-conformally covariantly-looking results. Eventually, they were able to show that, indeed, the conformal invariance was preserved on the level of one-loop divergent effective action.

As emphasized already a few times, it seems that the authors of [96] assumed what they wanted to prove by explicit computation, that is, that the conformal symmetry is not violated on the first-loop quantum level. *A posteriori*, they proved that their initial assumption was indeed correct, but in this way they entered into some kind of logical vicious circle. Somehow, we need an argument that does not rely on the presence of conformal invariance before it is explicitly checked that it is there. This argument is given in the form of analysis of the conformal anomaly at the one-loop level, that the same authors also discussed in later publications. We will discuss it also below. Here, we want to remark that all the considerations of the presence or absence of the conformal symmetry were considered so far only on the level of the divergent effective action, and the issue of conformal anomaly (CA) and its form were completely neglected for these aspects. So far, we have just analyzed whether the counterterms are conformally invariant, while, as emphasized earlier, the mere existence of them breaks full conformal symmetry due to presence of non-vanishing beta functions for couplings, and thus also of non-trivial RG flow and of non-trivial scale dependence of various correlation functions.

The results from the limit $\alpha_{R2} \rightarrow 0$ did naively contain the $R^2$ term, which is dangerous for full local conformal symmetry, but these were only incomplete results since additional care had to be given to them. By performing final conformal transformations, Fradkin and Tseytlin were able to reduce to zero such non-conformal counterterms; however, the parameter of such a transformation was expressed as highly non-local function in (41). They could still perform the conformal transformations on the form of counterterms as the last operation, because the symmetry they acknowledged earlier had to be present also on the quantum level since it was present also on the classical level before the quantization procedure. Implicitly, they assumed that the quantization act does not destroy conformal symmetry. They just used the freedom to select a conformal gauge, although one has a strong impression that their approach could be more clear since they knew only *a posteriori* that local conformal symmetry must have been present on the level of the first quantum loop in this theory. The fact that the solution for $\Omega$ contains this apparent non-locality in local Weyl conformal gravity in four-dimensional QFT framework could be quite disappointing, and one could wonder whether it is possible to perform all such procedures better and in a more clean way. Moreover, the parameter $\Omega$ must be always spacetime-dependent since constant values do not change the form of the Ricci scalar $R$ according to (36), because we always have that $\Box\Omega = 0$ for $\Omega = $ const.

When one is finished with the dangerous conformal symmetry-violating $R^2$ term in UV-divergences, then this is not the end of the story because, as authors say, the contribution of two scalar degrees of freedom has to be subtracted from the results in (38) without $R^2$ term there. These are assumed to be two minimally coupled to gravity scalar modes on a general background. At the end, the form of UV-divergences in Weyl conformal gravity in $d = 4$ spacetime dimensions regularized in the DIMREG scheme reads,

$$\Gamma_{\mathrm{div}}^{1-\mathrm{loop}} = -\frac{1}{(d-4)(4\pi)^2} \int d^4x \sqrt{|g|} \left( \frac{199}{30} C^2 - \frac{87}{20} \mathrm{GB} \right). \tag{42}$$

The results in the formula above should be understood as final ones. No additional conformal transformation or subtraction of some contributions should be performed on them.

One also wonders whether the final conformal transformation nullifying the $R^2$ term can be reversed. It is easily acceptable that one obtains as the result of conformal trans-

formation that $R \neq 0 \rightarrow R' = 0$. However, here is visible apparent irreversibility of the conformal transformation because one does not obtain anything from $R' = 0$ as the starting point, since zero transforms to zero under homogeneous transformations. Of course, this is only apparent since by inverse conformal transformations which are achieved with $\Omega^{-1}$ parameter, one obtains from $R' = 0$ back to the case $R'' = R \neq 0$, or one can even perform a new transformation with $\Omega \neq 1$, and then from $R' = 0$ one will find oneself in the spacetime where $R'' \neq 0$. One, therefore, has a strange impression that the power of conformal transformations in a theory with conformal symmetry on the quantum level was only used instrumentally by authors of [96] to reach the assumed goals and, for example, after doing this, one is forbidden to use it anymore, or to use it for reversing its effects. Of course, such a situation with understanding of the symmetry in the theory is not satisfactory, and one asks for a natural formalism which makes such transformations obvious, and all of them are allowed to be performed provided the symmetry is realized there on any level of the theory.

In other words, we can say that the $R^2$ term in UV-divergences of the theory is conformal gauge-dependent, that is, by a choice of some conformal gauge one can eliminate it but *a priori* it is there. We choose a conformal gauge by performing arbitrary conformal transformation in the theory. Is the $R^2$ term there or not?—This statement is not universal and it is actually gauge-dependent in conformal gravity. The value of the $R^2$ beta function is not universal either. Therefore, in the description preserving conformal symmetry we shall not speak at all about this term, and the issue related to its presence or non-existence is not a physical issue, it is just another example of the gauge artifact statement, here due to conformal gauge.

If one looks at the difference between formulas (42) and (38) (the last one obtained as the limit in the case $\alpha_{R2} = 0$), one sees that the correcting contribution of two scalars is essential and that this together with the resolution of the problem of the $R^2$ counterterm signifies why the simple naive limit $\alpha_{R2} \rightarrow 0$ does not work for the case of quantum conformal gravity. These are all post-computational procedures that one has to perform after the naive limit $\alpha_{R2} \rightarrow 0$ is taken. They constitute the theoretical reasons (or the problems within the formalisms of simple QFT of gravitational interactions with only diffeomorphisms symmetries) why the limits in question are discontinuous and why the case of conformal gravity is very special one.

*5.3. Subtraction of Two Scalars*

There could be various attempts to explain the role of the mysterious two scalar degrees of freedom that have to be subtracted for the last corrections to the theory. The possible questions are why they have to be subtracted, not added, why they are two, and why scalar degrees of freedom, and not, for example, one massless vector (spin-1) or tensor (spin-2), each of the last two coming with two helicity states as well. Finally, one could ask, accepting that they are scalars, why they have to be minimally coupled to a curved gravitational background, and not conformally coupled to gravity. The last option looks to be the most reasonable one, especially in the framework of conformal gravity, where conformal symmetry of the whole physical system (gravity + matter) on the quantum level is the thing we care so much about here.

A possible explanation of the problem with two scalars may come from analyzing the spectrum. Exactly two degrees of freedom are the difference between eight and six perturbative degrees of freedom (as analyzed around flat spacetime background), respectively, for generic HD gravitational theories and for Weyl conformal gravity. Hence, if one wants to obtain six degrees of freedom one would have to subtract two degrees of freedom from all degrees of freedom present in HD gravity with both $C^2$ and $R^2$ terms. The arithmetic here agrees. However, there could be some problems with this interpretation. First, as we discussed before, this way of counting degrees of freedom is not completely unambiguous and one may be tempted to assume number of six degrees of freedom also in generic HD gravitational theory without the Einstein–Hilbert term added. Secondly, when one looks

at the spectrum of generic HD theory, one does not see that there are two more scalar degrees of freedom compared to Weyl gravity. One sees only one physical scalar degree of freedom in generic HD theory (with standard counting to give $8 = 2 + 5 + 1$ degrees of freedom), while in conformal gravity, there are no scalars at all[10]. The actual reshuffling of degrees of freedom (from eight to six) is intricately more complicated: first, one must subtract one scalar d.o.f. and then later divide five d.o.f.s from massive spin-2 field into two helicities of massless spin-2 and another two helicity states of massless spin-1 (fields present in conformal gravity), and finally one has to remove the last remaining degree of freedom. This game with various Lorentz representations of perturbative degrees of freedom would suggest to try something with one scalar, massless vector or massive spin-2 mode. However, it is incorrect to add or subtract here, for example, the contribution to one-loop divergences of traces of a vector (spin-1) and the final results work only when precisely two scalars minimally coupled are subtracted. Hence, this reinterpretation of subtracting contribution of two scalars can be, at most, only viewed as a heuristic tentative explanation.

On the other hand, one could also come up with a different idea that these two scalars are conformal ghosts. This would explain why, at the end, Fradkin and Tseytlin had to take them into account when limiting to Weyl gravity (with enhanced conformal symmetry). These fields obviously were not present in generic HD gravities, but when one moves to conformal theory, one has to gauge-fix this symmetry and also supply proper ghosts (depending on the character of interactions, they have to be either FP ghosts of new local symmetry or third ghosts if HD terms are present, or both). It is true that the authors of [96] performed an additional simple conformal gauge-fixing, requiring vanishing of the trace $h = 0$. This performs the first part of the job of treating with due care the eminent conformal symmetry here. However, the next part is to deal with respective ghost fields of conformal symmetry, and, moreover, what is reassuring here is that since they are of Grassmannian nature (as all quantization ghosts are), then to add their contribution means effectively to subtract the contribution of "normal" particles. Hence, this is consistent with the idea that to reach conformal gravity, one has to add contribution of conformal ghosts, or, in a sense, subtract contribution of normal particles.

As it could be explained thoroughly in the next publications on this matter, these conformal ghost fields are here in four-dimensional conformal gravity of the type of third ghosts exclusively, namely, they are not FP conformal ghosts. However, what is more important here is that they are of scalar character, and they do not carry any Lorentz index. Moreover, as will be made clear in the next section, they are just minimally (covariantly, similar to in GR) coupled to the background and not conformally coupled, since such is the feature of the FP quantization procedure. This fits very well with the description of subtraction of two scalars; however, again, there could be some problems with such an interpretation. First, ghosts used during the covariant quantization procedure (that is, FP ghosts or third ghosts) are typically not counted for active perturbative degrees of freedom around a given background, hence they would not entail any change in the number of degrees of freedom between the two theories (limiting HD gravity and conformal gravity). This is in clear contradiction with the counting suggesting that the change is $8 \to 6$, while it is also in perfect agreement with other counting, where we have advocated before that there is no change and we have $6 \to 6$. One cannot also say that potential additional two degrees of freedom, as present in the standard counting of degrees of freedom of HD gravity, are these two conformal ghost modes (counted negatively). The ambiguity and the potential problem with this interpretation remain.

However, after all, the role of such ghost contributions (both whether these are FP or third ghosts) is not to change the number of perturbative active (physical to some extent) degrees of freedom, but to provide the correct expression for the partition function. For example, from such an expression, one could, at the one-loop level when expressed via determinants of some differential operators, obtain the one-loop partition function and, finally, read from it the number of degrees of freedom. This number typically agrees with

the number obtained via counting of constraints and other similar canonical methods (and, for example, it gives eight degrees of freedom in scale-invariant HD gravity). However, on the level of such an expression written using determinants in various subspaces, it is not evident to understand the origin of various factors and functional traces there. As emphasized and, for example, explained in the case of conformal gravity, one can find there some various auxiliary elements of the theory used for quantization and for writing path integrals, such as contributions from Jacobians of change of field variables under path integrals, from gauge-fixing functionals, and also from FP and third ghost determinants. One does not find there contributions clearly assigned to modes appearing in the perturbative spectrum of the theory around flat spacetime, such as no clear origin with massless spin-2 or spin-1 fields. Eventually, from one-loop partition function, one finds the correct total number of degrees of freedom (including also these spin-2 and spin-1 massless states in conformal gravity). However, eventually, there is no sign that this counting manifestly takes FP ghosts or Jacobians into account. It only has to take them in intermediate levels of computation for the whole consistency of the procedure.

To answer the question of why there are two scalars, one has to recall a few facts. First, it is well known that the third ghost is just one single real field. This would already look like a contradiction to our counting, since we stated that there are two scalar contributions that need to be subtracted, but secondly, one has to pay enough attention to the fact that the mentioned contributions of minimally coupled real scalars are understood as coming from the following two-derivative scalar action coupled to curved spacetime background:

$$S_{\mathrm{sc}} = \frac{1}{2} \int d^4x \sqrt{|g|} g^{\mu\nu} (\partial_\mu \phi)(\partial_\nu \phi) = -\frac{1}{2} \int d^4x \sqrt{|g|} \phi \Box \phi, \tag{43}$$

where we do not write any scalar potential term $V$ (containing mass terms or self-interactions) since it is well known that the terms in $V$ do not at all contribute to divergence proportional to $R^2$ and GB term. This is undoubtedly true at the one-loop order and for dimension-four counterterms in the effective action, which could receive contributions only from dimensionless couplings, but, for example, here they do not receive any contribution from quartic self-interactions of scalars. Only the form of the kinetic operator (precisely, how many derivatives there are) and the multiplicities of scalars (how many scalars there are) matter for this computation. There is a slight dependence of the contribution to the $C^2$ counterterm on the non-minimal coupling $\xi$ with gravity (that is, with a possible non-minimal term $\xi R \phi^2$, which is so-called coupling to a scalar curvature of spacetime), but these are all neglected for our purposes. One could say that, at the end, our model is very poor, since it describes here two massless scalar fields with no interactions, no self-interactions, and even without mass parameters, just coupled simplistically and minimally to the background geometry. Even the normalization of the scalar field here is not an issue for UV-divergences, but here we stick to the canonical one.

One has to compare the above with the contribution from the $G_c$ matrix. To make the story short, the suitable operator in $d = 4$ is of the form $G_c = \Box^2$ and this acts between two real scalar third ghosts for conformal symmetry $b_c$. Hence, due to fractionalization property of functional traces of a logarithm of such an operator as $G_c$, one sees that on the level of the trace of the logarithm, its contribution precisely agrees with two contributions from one scalar box $\Box$ in a simple scalar field representation in the last part of the formula (43). This explains the presence of two scalars. In more generality, one could say that, in general, even dimensionality of spacetime $d$, the number of simple two-derivative scalars, each with action in (43), the contribution of which have to be subtracted, has to be equal to half of the dimension $\frac{1}{2}d$. This assertion can place our theoretical explanation in verification, for example, in six-dimensional conformal gravity models.

Within this second interpretation, we have a confirmation for why these two contributions have to be subtracted, but not added, and why they are scalar degrees of freedom, and not massless vectors or symmetric rank-2 tensors. We also see why they are minimally coupled to GR-covariant background and why they are two scalars. However, as remarked

before, the whole existence or need for third ghosts (both of diffeomorphism symmetry or of local conformal symmetry) depends on the choice of the background. As we emphasized previously with some clever choice of partially covariant weighting functional $G_c$, their contribution can be safely eliminated and they can be forgotten on the quantum level. Such is, for example, the situation around flat spacetime, when, in the gauge-fixing functional $G_c$, one uses only partial derivatives, and from them builds scalar d'Alembertian operator $\partial^2$, and on the contrary, one does not use background-covariant derivatives from GR to construct a proper GR-covariant box operator $\Box = \nabla^2$. Then, in such circumstances, there are no conformal third ghosts at all and so their contribution also does not exist and then this interpretation breaks down.

One could try to rescue this idea that the two scalars are third ghosts for conformal symmetry by claiming that for GR-covariant results of one-loop divergences obtained within some covariant framework, such as within the method of covariant Barvinsky–Vilkovisky traces, one ought to use only the form of the weighting functional $G_c$, which is GR-covariant, so it must be built using the covariant d'Alembertian operator $\Box$. Otherwise, one could be afraid of losing the general covariance of the results at the end, but they are guaranteed to be provided such that in the whole process of computation of them there are no steps which explicitly and manifestly violate the general covariance. Using only partially covariant $G_c$ functional (or in a sense forgetting about the third ghosts contributions) is not in line with such a formalism, and the results obtained by this oversight

are, of course, incorrect. Hence, one can conclude that third ghosts must be remembered in all GR-covariant formalisms of computation, while one can still safely forget about them around flat spacetime and when working with Feynman diagrams.

However, here one must remember that the original computation due to Fradkin and Tseytlin was performed just in manifestly covariant framework and using background field method (BFM), when they sum various contributions and they do not use non-gauge-covariant results from some partial resummation of Feynman diagrams. It is well known that the two methods at the end should agree for final results for UV-divergences, which are proved to be expressed via gauge-invariant and gauge-fixing-independent counterterms. One must remark that in the covariant BFM formalism, as the Barvinsky–Vilkovisky (BV) trace technology is, the partial results, although looking generally covariant (so in this sense they are better looking than results of some subset of Feynman diagram contributions, which are not gauge-invariant), are without any sense if considered separately. Only their total sum has a clear physical meaning. Therefore, one cannot say with a good meaning, for example, what the generally covariantly written and unambiguous contribution of FP ghosts (or third ghosts) is to total divergences. Namely, it is true that they contribute and one usually should not forget about them, but their contribution alone is not physical. One should not be deluded by intermediate results, which look covariant, and would invite an interpretation of such partial results. Therefore, one must remember third ghosts for local conformal symmetry, but, again, their contribution alone is not very meaningful.

It is still a very interesting fact that their contribution (with all the provisos mentioned above) precisely agrees with the discontinuity between conformal gravity and a four-dimensional HD gravity after the limit $\alpha_{R2} \to 0$, thus realizing the subtraction of two scalar degrees of freedom minimally coupled to background geometry. It remains yet to be seen if such a discontinuity has any physical meaning, or it is just an artifact of our clumsy way of approaching conformal gravity from generic HD gravity. Therefore, the interpretation of two scalars as two third ghosts for conformal symmetry in Weyl gravity is quite a plausible one, since it defends itself quite well, and certainly it is worthy studying further. For this, one should also better understand the issues related to conformal third ghosts.

Instead, here we summarize what is known about the contribution of two scalars to UV-divergences. This does not presuppose that these scalar fields are real or physical or whether they really live on a curved spacetime background. Nonetheless, their contribution can be singled out (with all the provisos above). Namely, two minimally coupled scalar

fields (each with one degree of freedom and with a two-derivative action from (43)) on a curved general background contribute twice the following contribution to UV-divergences,

$$\Gamma_{\mathrm{div,\,sc}}^{1-\mathrm{loop}} = -\frac{1}{(d-4)(4\pi)^2}\int d^4x\sqrt{|g|}\left(\frac{1}{120}C^2 + \frac{1}{72}R^2 - \frac{1}{360}\mathrm{GB} + \frac{1}{30}\Box R\right). \quad (44)$$

The last term above, proportional to $\Box R$, was written just for definiteness and we do not use it in the final main computation. The contribution of two scalars in (42) can also be studied in the $R^2$ sector of UV-divergences, although the final result is non-universal and it leads to a conformal gauge-dependent coefficient $\frac{1}{9}R^2$, which was justifiably neglected when writing the final formula (42).

*5.4. Relation to Conformal Anomaly*

When one speaks about UV-divergences at the one-loop level, one should not forget to relate them to the celebrated conformal anomaly. Basically, on the quantum level where there are UV-infinities and also resulting beta functions of couplings, there is an RG flow, and Green functions do not show perfect scale invariance. They depend on energy scale, and this dependence is dictated by RG phenomena. This means that inevitably in such situation, the conformal symmetry of the matter model (global group, as we discussed in Section 2.1) is broken and we do not have a globally conformally invariant model on flat spacetime. This is also confirmed by analyzing conformal anomaly of global conformal symmetry in the setup of a curved geometric background (as explained in Section 2.2). Then, the global group of scale transformations from GR in some matter models (which were globally conformally invariant) may be broken on the quantum level when one couples these matter theories to a non-trivial gravitational background. As for global anomaly of conformal symmetry, this is not a big deal in matter theories, where conformality on the classical level might be just a lucky coincidence, and then one does not view the effects of this anomaly as disastrous for the original matter theory. Simply, on the classical (tree-) level before the quantization, the conformal symmetry was present, but just after quantization of matter fields and matter symmetries (which could be also local internal symmetries) due to quantum effects, this symmetry is not there anymore and it does not constrain quantum dynamics anymore. One loses some power and simplicity of the theory but then the quantum physics enters and all correlations have to be tediously computed according to rules of QFT (in distinction from rules of CFT). One cannot use the rigidity of constraints placed on Green functions due to CFT models anymore, since one is out of their realm.

Technically speaking, the conformal anomaly (CA) understood as an anomaly for global conformal symmetry is equivalent to a trace anomaly. The last one signifies that the trace of the energy-momentum tensor of the matter theory does not vanish, although the theory on the classical level possessed scale invariance or even conformal invariance (before this was coupled to gravity). These last two requirements on the classical level imply that the classical EMT[11] is trace-free ($T = 0$) upon using EOM from the matter sector, and this is completely tantamount to invariance of the theory under infinitesimal conformal transformations understood in the framework of GR (as they were studied in Section 2.2). Therefore, we end up in the situation that on the quantum level, due to physical quantum effects (but not due to an act of some uncareful quantization), one finds that the trace of EMT on-shell is not zero anymore. For this, one has to use the quantum matter equation of motion obtained from the full quantum effective action restricted to the matter sector. One can specify the situation at the perturbative one-loop and then one correctly expects that this violation of tracelessness condition (thus, violation of scale invariance) is proportional to beta functions of the theory, which in turn are proportional to UV-divergences. Such a simplistic identification is no longer true at higher loop orders, but we can restrict all of the below analysis to this case only.

To summarize, there is an anomaly because some condition known from the classical theory (here, that $T = 0$) is not present on the quantum level of the theory anymore. If some symmetry is on the classical and was expected to be a symmetry of the theory at

all conditions, then its lack on the quantum level is a true anomaly in a physical sense, and this tells us that maybe we had a wrong expectation about the theory, and that theory with this symmetry is simply anomalous and very sensitive to the quantum effects, or that the original matter theory with conformal symmetry was not in a symmetric enough situation to protect the conformal symmetry from violation at the quantum level. Since, in the matter models, this conformal symmetry is not in a gauged form, there actually is not a big problem with such an anomaly. One has to simply accept that symmetries on the quantum level of the model are in the poorer form, and, for example, conformal symmetry is not realized in the quantum dynamics.

Still, in the matter models, but coupled to an external geometry, one expects that the lack of conformal symmetry on the quantum level manifests itself as the presence of non-trivial counterterms. Here, we can concentrate on UV-divergences in the gravitational sector, but one can also consider them in the matter part. For what we are going to say, this last restriction will be completely sufficient. Therefore, we can define that conformal anomaly $\mathcal{A}_c$ is exactly proportional to the $b_4$ coefficient of UV-divergences of the theory at the one-loop of accuracy. Moreover, one can view it as proper trace of the quantum EMT defined from the effective action, that is, we have

$$T = g^{\mu\nu} T_{\mu\nu} = -\frac{2}{\sqrt{|g|}} g^{\mu\nu} \frac{\delta\Gamma_{\text{div}}}{\delta g_{\mu\nu}} = \frac{1}{(4\pi)^2} b_4. \tag{45}$$

It is important to realize that in the above expression, the divergent part of the effective action is taken in full generality as off-shell, so without using quantum (or even classical) EOM. For matter models, the divergent effective action there $\Gamma_{\text{div}}$ is with all contributions when the matter fields are taken quantum (so they run in the loops of Feynman diagrams), while the gravity is treated as a classical external field to which the quantum matter system is coupled (so graviton lines can be only external lines of the diagrams). Moreover, in the above formula (45), a regulator which was used to isolate UV-divergences was already extracted, so the expression for $b_4$ (and also equivalently for $T$) is finite and not UV-divergent.

The $b_4$ coefficient of divergences is, of course, special to $d = 4$ dimensions because then counterterms have the energy dimension equal to the dimensionality of spacetime and couplings in front of them are dimensionless, so naively we already have classical scale invariance. Its relation to the divergent part of the one-loop effective action (part with classical scale invariance of the action) reads,

$$\Gamma_{\text{div}} = \frac{1}{(d-4)(4\pi)^2} \int d^4x \sqrt{|g|} b_4. \tag{46}$$

In the case of only gravitational counterterms that we are interested in, this coefficient is related to the beta functions of couplings in the following way:

$$b_4 = \beta_C C^2 + \beta_R R^2 + \beta_{\text{GB}} \text{GB} + \beta_{\Box R} \Box R. \tag{47}$$

One notices a few interesting facts about this expression for anomaly. First, the coefficient $\beta_R$ is not universal since it is conformal gauge-dependent (for example, in the background conformal gauge, its value is zero). Secondly, the beta function of the total derivative term $\beta_{\Box R}$ is known to be ambiguous on the level of the trace of EMT, but not on the level of the $b_4$ coefficient. In the results we can forget about these last two terms and consider only universal contributions to two beta functions: $\beta_C$ and $\beta_{\text{GB}}$. Once again, we remind the reader that the presence of CA is not problematic for matter models, where this symmetry does not show up in the gauged form on the quantum level. One really sees, unambiguously, the presence of CA when, off-shell, the expression for it is not proportional to matter EOM.

One can see the consequences of CA for the quantum matter theory not only on the level of ultraviolet divergent expressions in correlation functions, but also in finite parts of

these correlators on the quantum level computed to a given accuracy. For example, due to RG flow phenomena and within RG formalism (which basically tells in the simplest way that physical results should be independent on the arbitrary renormalization energy scale), one finds that the finite terms in the semi-classical general off-shell action for gravity (when quantum matter fields are integrated over) take the following form:

$$\Gamma_{\text{fin}} \supset \int d^4 x \sqrt{|g|} \left( \beta_C C \log \frac{\Box}{\mu^2} C + \beta_{\text{GB}} \text{GB}_{\log \Box / \mu^2} \right), \tag{48}$$

where, for simplicity, we wrote only contributions which are universal on the level of the trace of EMT $T$ related to the $b_4$ coefficient. The non-local insertion of the logarithm of the GR-covariant box operator is both present between two Weyl tensors and also between Gauss–Bonnet invariant. The last one explicitly rewritten takes the form

$$\text{GB}_{\log \Box / \mu^2} = R_{\mu\nu\rho\sigma} \log \frac{\Box}{\mu^2} R^{\mu\nu\rho\sigma} - 4 R_{\mu\nu} \log \frac{\Box}{\mu^2} R^{\mu\nu} + R \log \frac{\Box}{\mu^2} R, \tag{49}$$

which is in the direct analogy with the formula (31). One sees that these contributions in (48) explicitly break conformal symmetry, despite that on the level of divergent action, so without non-local logarithms of the box operator, they entail completely conformally invariant counterterms. This shows that in such a setting, the conformal symmetry is explicitly broken and it is no longer there on the quantum level. As one confirms in (48), the expressions vanish when the corresponding beta functions vanish. Only there we are sure of quantum conformality present in some matter models. In the expressions in (48), we see highly non-local functions, namely, of the logarithm of the covariant box operator. Their origin is due to RG invariance of the total effective action when the running of couplings is also taken into account and where $\mu$ is a physical scale of renormalization which cancels with the $\mu$-dependence of running coupling parameters for any physical observable computed within the given model.

Thus far, we have only discussed some issues related to conformal anomaly when gravity was the external dynamical classical field. When we use the same procedure towards quantum gravity when it is both dynamical and quantum, and, for example, propagates inside quantum loops (where there is finally graviton's propagator), and gravitons can be on internal lines of some Feynman diagrams, then in conformal gravity (so, the theory with local conformal symmetry), one has to be very careful since now this symmetry is in the gauged form. If one sees an anomaly, this might be a sign of a just creation of pathological, sick theory. This issue of how CA in conformal gravity may create disastrous effects was discussed at length in various previous works [36–39]. We will also discuss these issues and potential resolution of them at full length elsewhere.

Now, if the conformal symmetry is in the gauged (local) form, then one can adapt the discussion which was presented above to the total system composed of matter and gravity when both components are quantum and dynamical. The conformal symmetry was the defining local symmetry on the classical level of gravitational theory, hence the issue with potential conformal anomaly for it is very important and crucial to resolve satisfactorily. Moreover, local conformal symmetry is instrumental in defining the spectrum of the theory, for example, around flat spacetime, so any change with this would ruin the spectrum and classification of irreducible Lorentz representations of modes there. It would also destroy the counting of perturbative degrees of freedom of the theory. All these problems show in other disguise as violation of unitarity, not due to the HD character of gravitational theory, but due to destroying one of the local gauge symmetries, here in this case of local conformal symmetry. We will not comment on these perennial issues here, but we will refer to the vast literature on this topic.

In the case of a total gravitational system where matter is coupled to gravity, one has to analyze the total EMT of the full system. As is well known, it is composed of two pieces—gravitational and matter parts. One also notices that following Weinberg's definition, the



total EMT is zero on-shell as a consequence of invariance of the total action (gravitational part and coupled matter part) under GCT. Thus, on-shell we do not find any problem with CA in the total system. This remark applies both to the classical and quantum level, since in the last case, one can, for example, compute the effective EMT from the full quantum effective action functional, and for this argument to work, the gravitational theory or matter part alone do not have to be even classically conformally symmetric. This is indeed a very robust argument valid in any diffeomorphism invariant theory, which also preserves this symmetry on the quantum level (so, under the condition that there is no quantum anomaly of diffeomorphism symmetry). In any such theories on-shell, we find that the trace anomaly $T$ vanishes. However, as usual in QFT, the situation off-shell is also very important, and, actually, in these circumstances one can again relate this trace $T = \mathcal{A}_c$ to the coefficient $b_4$ of divergences according to the last formulas (45)–(47). These formulas, also in the more general case, with quantum gravitational part remain valid. Of course, the respective beta functions now receive contributions from both quantum gravitational interactions and the quantum matter sector.

In the total system interacting gravitationally, the issue of detecting physical consequences from the expressions for $b_4$ is more complicated than just a case of simple matter theory. First, one knows that even on a classical level there exist some matter models for which the trace $T$ does not vanish classically off-shell but only using matter EOM. Now, with gravity, we have this twist where $T = 0$ on-shell as a kind of tautology. One must definitely analyze the situation quantum off-shell, but even if one finds there that $T \neq 0$, this does not necessarily imply that there is a conformal anomaly in the total system. Sometimes one finds for the total $T$ off-shell an expression which does vanish, then one is sure of the absence of CA, or in the gravitational context, if the expression for it is proportional neither to the gravitational EOM nor to the EOM from the matter sector. These EOM should be derived from the respective quantum EOM originated from quantum effective action. This is a conformal Noether identity for the total system (matter + gravity).

One could also in parallel analyze the situation in six dimensions since then the space for all terms in conformal gravity is bigger. It is interesting viewing the situation with divergences in six-dimensional conformal gravity of the type $C \Box C + \ldots$ which still possesses the propagator around flat spacetime. We expect only three conformally invariant counterterms, $C \Box C + \ldots$, $(C^3)_1$, and $(C^3)_2$, out of 10 possible terms in the action with six dimension operators [98,99]. There, we also expect that the conformal anomaly $\mathcal{A}_c$ is described in a conformally covariant way such that $\mathcal{A}_c = b_6$ and that only three terms, $C \Box C + \ldots$, $(C^3)_1$, and $(C^3)_2$, out of 10 possible terms in the action with six dimension operators appear there. What about the remaining seven terms? If all this formalism is correct they should be made to vanish by just one conformal transformation in six dimensions with just one parameter $\Omega(x)$. This is quite improbable in a general situation to remove six counterterms (although they are related) by just one conformal transformation, but it should be true in a general case when the reasoning with conformal anomaly is correct. The UV-divergences after conformal transformations should be described by conformally invariant terms only in $\mathcal{A}_c$, according to the general wisdom that terms in conformal anomalies are written in a way that preserves conformal symmetry (despite that they are in the anomaly heralding the disastrous breaking of this symmetry).

*5.5. Summary of the Method*

The results of all these above procedures, although maybe not deeply justified, revealed to be true for Weyl conformal gravity, as it was checked after using other methods of computation. The coefficient of the $R^2$ divergence indeed vanishes at the level of the first loop, and the two other coefficients were correctly obtained in the limit $\alpha_{R2} \to 0$: for $C^2$ and for the Gauss–Bonnet divergences. This procedure was justified to be used and to be a correct one. The potential general problem could be that only diffeomorphism symmetry was treated with due care since quantization and regularization procedures were performed in a way to preserve it. The conformal symmetry was restored by hand

using final conformal transformations, where one actually admits that this symmetry was always there, even on the quantum level, but maybe in its hidden form. Of course, one could act much better and treat it well from the beginning, and then at the end, one would not have to perform correcting conformal transformation. However, even if we want to perform this rectification procedure, there would be a clear justification that this is possible since it would be obvious in a more manifest formalism that the conformal symmetry is present on the quantum level. One would not have to perform these compensating final transformations with parameters which are apparently strongly non-local functions of spacetime points.

Finally, this procedure, as employed confidently by the authors, recovers local conformal symmetry on the quantum level, although there were some hidden assumptions and not *a priori* justified steps in its rederivation. One could ask how such procedure can be systematized, if at all, which transformations we should perform at the end, and whether this works also to higher loop level, or only at the level of the first loop. One may wish to work all the time from the start within the formalism that preserves conformal symmetry in the framework of covariant quantization of theories with local gauge symmetries. This is exactly what we plan to explain in the future sequel article of this paper.

## 6. Conclusions

In this introductory review article, we tried to present and extensively motivate the need for a new quantization procedure. This method should be designed to specially treat local conformal symmetry when this is present in gravitational interactions introduced in the same way as Weyl did in 1918. Local conformal symmetry is very special, as we emphasized in the introduction to this paper and also in the motivations.

We discuss how to view it as a gauge symmetry both on the flat spacetime and also on curved spacetime framework. The latter is generally best suited to study fields and their transformations, as in the proper GR framework. Keeping this in mind, we later viewed conformal symmetry as a particular example of the general gauge theories. However, compared to other gauge symmetries, in the minimal version, conformal symmetry comes with some differences. It does not come with its own potential field, but it just uses the metric, somehow borrowing it from the diffeomorphism group. This relation is another emanation of the fact that the full conformal group is not a direct product of the Poincaré group and the proper conformal factor (which consists of five generators in $d = 4$ spacetime dimensions). Moreover, we emphasize that the transformation law for the metric field, which could be treated similar to a charged matter for conformal symmetry, does not contain any differential operators, and this is just an algebraic operation of multiplication. In this way, the conformal gauge theory differs from any other local gauge theory with standard potential fields, wherein the generators of infinitesimal gauge transformations are generically differential operators.

This last observation has profound implications for the quantization program of conformal symmetry in conformal gravity. Namely, this implies that in simple conformal gauges, we do not need to supply Faddeev–Popov ghost fields for conformal symmetry, since they can always be algebraically eliminated. However, one always has to additionally fix the gauge, remembering the local conformal symmetry with its special gauge-fixing conditions. The simplest choice is to place some constraints on the trace of the metric fluctuation field $h = g^{\mu\nu}h_{\mu\nu}$. The easiest choice is to just place some algebraic condition, the more complicated conformal gauge choice is to place some differential condition here [100]. Irrespective of which choice is selected on a general background for the consistency of the whole quantization approach, one must add to the spectrum one real minimally gravitationally coupled scalar field, which here plays the role of the third ghost field for conformal symmetry. This third ghost comes naturally with a four-derivative kinetic term. Equivalently, one can add two real scalar fields minimally coupled and enjoying two-derivative dynamics. The need for this additional third ghost scalar field is because of the higher-derivative nature of conformal gravity in $d = 4$ (actually, this is true in any $d > 2$). We also

emphasize that these conformal third ghosts are different to the usual third ghost fields related to diffeomorphisms, which are also present in general theories (not necessarily conformally invariant) with higher derivatives on a general background. Since these third ghost fields couple only to the gravitational background, then if one studies Feynman rules around flat spacetime (gravitational vacuum configuration), then one can safely forget about them. They are needed for propagation of modes around non-trivial spacetime configurations or for necessarily and inherently background-independent approaches to QG, such as, for example, in Barvinsky–Vilkovisky (BV) [101] trace technology, which is a generalization of the Schwinger–DeWitt covariant methods of computing the effective action at the one-loop level around general curved backgrounds.

These conformal third ghost fields also played an essential role in our explanation for the one-loop results correctly obtained for the first time by Fradkin and Tseytlin regarding the UV divergences in Weyl gravity [96]. Their method, although looking a bit ad hoc, was correct, and we found a justification for why this was so. From the generic results that one obtains in Stelle quadratic theory with two independent terms, $\omega_R R^2$ and $\omega_C C^2$, one first takes the limit $\omega_R \to 0$, obtaining some finite results. However, they are not the final ones because of the mentioned Veltman-like discontinuities in conformal gravity. Their cure is to subtract the contribution of two real scalars minimally coupled. They worked in the fully covariant framework of BV methods, hence it was natural for them to work on a general curved background. The explanation why these are two scalar, real, and minimally coupled conformal ghosts was found by identifying them as the third conformal ghost fields, which are new fields in the here-presented proper conformal quantization approach to conformal gravity, and they have to be included on general backgrounds. This proves that our framework of quantization specialized to the cases of local conformal symmetry in the gravitational setup already brings correct and valid results.

Having constructed a detailed framework for quantization which has breaking of conformal symmetry under full control, we can now use this approach to write with full confidence the Feynman rules of the perturbative theory that we can use for computation, for example, at the two-loop level. Such computations are very interesting, though very difficult too, in pure conformal gravity to check the consistency of conformal symmetry on this loop level. They are also very important for the further developments in FT conformal supergravity theories. As primary areas of application of our results, we mentioned the two above. In pure conformal gravity in $d = 4$, we could investigate the situation at higher loop orders and decide the fate of conformal symmetry on the quantum level, the presence of conformal anomaly, and dangerous $R^2$ terms. In this model, we need to exert special care to check whether the conformal symmetry is finally dramatically violated by quantum corrections and not by the mere quantization method or procedure. For example, having the well-defined formalism of Feynman rules in conformal gravity around a flat background, one could extend the computation of scattering amplitudes from tree-level [102] (where it coincides with the results of Einstein–Hilbert gravity) to higher loop orders, the results of which will surely be very interesting regarding the unitarity bound of gravitons' scattering.

Instead, in the framework of $\mathcal{N} = 4$ conformal FT supergravity, we know that the symmetry is fully preserved on the quantum loop level. Therefore, one is required and tempted here to work in the formalism, which preserves this symmetry on the quantum level, or at least which performs the gauge-fixing of this symmetry in a fully controllable way. Similarly to how one uses superfield formalism and supersymmetric gauge-fixing of local symmetry to preserve this symmetry in local supergravity models, in the framework of FT supergravity, we shall implement the same with the conformal part of the full superconformal algebra of symmetries. As it is well known, to perform a real quantum computation at some loop orders, one really has to fix the gauge for all local symmetries, otherwise one meets a degeneracy problem of the kinetic terms for small fluctuations around the flat spacetime, and then the perturbative propagator cannot be defined. To avoid these unfortunate circumstances, one adds gauge-fixings and breaks all local symmetries, but in the controllable manner. This is all quantified by the presence of gauge-fixing parameters.

In the final results, for example, for *S*-matrix elements, one expects complete independence of these spurious gauge-fixing parameters. Then, this means that the symmetry is restored in these final results and the effects of intentional breaking are fully under control. When one applies our results regarding the correct quantization method, then the computation using, for example, Feynman diagrams and around flat spacetime can be brought to the very end. Moreover, one is then sure that the conformal symmetry is not touched and has its powerful impact and the constraining power on the final results. This is what is expected in the highly symmetric framework of $\mathcal{N} = 4$ FT conformal supergravity and regarding perturbative results in this theory.

Of course, there are also other possible applications of the framework for conformal quantization that we have presented here besides the two mentioned above. One can use it also in the BV framework at the one-loop level to perform honest and fully controlled computation of UV divergences in conformal gravity without any ad hoc unjustified methods or tricks. We have already performed such a computation and these results fully confirmed the ones obtained using standard non-conformally covariant techniques. We will report about the details of them elsewhere.

**Funding:** This research received no external funding.

**Institutional Review Board Statement:** Not applicable.

**Informed Consent Statement:** Not applicable.

**Data Availability Statement:** Not applicable.

**Acknowledgments:** We would like to thank I.L. Shapiro, P. Jizba, and L. Modesto for initial comments and encouragement regarding this work. Especially useful comments regarding the issues with conformal anomaly on the quantum level we owe to S. Giaccari. L.R. thanks the Department of Physics of the Federal University of Juiz de Fora for kind hospitality and FAPEMIG for a technical support. Finally, we would like to express our gratitude to the organizers of the TQTG'21 conference ("The Quantum & The Gravity") for accepting our talk proposal and for creating a stimulating environment for scientific online discussions.

**Conflicts of Interest:** The author declares no conflicts of interest.

## Notes

[1] The situation looks much more simple when one uses Dirac formalism and works in extended six-dimensional spacetime, although such a spacetime with the signature $(2,4)$ does not have a clear physical meaning. Moreover, here it can be understood as a useful, but only a theoretical, tool.

[2] One notices that the Minkowski metric $\eta_{\mu\nu}$ is completely unchanged under transformations from (1) (conformal rescalings), even with $\lambda(x)$ spacetime-dependent. The same is true also for SCTs acting on the metric of flat spacetime.

[3] One can note another interesting linguistic coincidence here, but this time in Polish language. The second morpheme "formal" in the word "conformal" (translated as Polish "konforemny") is related in Polish to the way that very symmetric regular polygons in plane geometry are named, originally known as "wielokąty foremne". Of course, such regular flat polygons have all the internal angles identical.

[4] But one sees a trivial thing here, that $g^{\mu\nu}g_{\mu\nu} = d$ is invariant under conformal transformations, stating obviously that the dimension of a spacetime is not modified by conformal transformations in (3). This may resemble an invariance in QED of $|\phi|^2 = \phi^\dagger\phi$, although the sense of conjugation is completely different here for real metrics.

[5] To avoid clash of notation with various curvature tensors, all denoted by the same letter "R" due to Riemann, Ricci, etc., we use the letter "R" in the Roman font to denote various infinitesimal generators of transformations.

[6] In actuality, it is completely opposite: for example, in standard Yang–Mills theories, perturbative FP ghosts are needed to save the unitarity of the perturbative covariant quantum non-Abelian gauge theory, as explained in [59]. The Feynman diagrams with them running inside loops are necessary to include for the whole consistency of the quantum theory.

[7] The same arguments correctly show that in the case of diffeomorphism symmetry, due to the fact that the infinitesimal generator of GCT comes with three indices, and we use a vector gauge-fixing $\chi_\mu$, the contraction in two pairs of indices results in only two indices left on the FP operator whose determinant can be easily defined and taken.

8     As introduced in Section 2, for the full quantum conformality, one must not have any UV-divergences. Hence, the developments above are only formal and the quantum effective action in theory with non-vanishing divergences is only superficially conformally invariant.

9     In principle, one could also add here a different term written as the Gauss–Bonnet density, but it is topological in $d = 4$ and its variation is a total derivative, hence vanishes under spacetime volume integral.

10    In actuality, to twist the story even more, we could add that proponents of counting of six degrees of freedom in massless scale-invariant HD gravity (but not in conformal gravity!) would see that there indeed are two perturbative scalar degrees of freedom (with spin-0).

11    The definition of the energy-momentum tensor (EMT) that one uses here is the unambiguous Hilbert definition as the variational derivative of the action with respect to metric fluctuations and properly de-densitized.

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
