# Peer review of "Introduction to Quantization of Conformal Gravity"

_universe, doi:10.3390/universe8040225_

Round 1

Reviewer 1 Report

The paper is devoted to a very interesting topic of quantising conformal gravity. However, the presentation needs to be better. First of all, there are certain problems of English language, like a constantly used unnatural collocation "no any". But also the very style of presentation can be described as "many words with very few formulae".

The Author discusses in great detail the very standard things about quantising gauge theories: gauge-fixing terms, Faddeev-Popov method, its ghosts. However, he does that rather abstractly, without explicit examples. For experts in the field, many parts are full of just obvious statements. At the same time, for non-experts it most probably won't be easily understandable at all. What is the expected audience for this paper?

A similar problem is there in describing the structure of the standard conformal group. And not only. Many important things are introduced almost solely in words. At the same time, for example, the formula (41) gives an absolutely obvious and standard rewriting of the scalar kinetic term.

At the same time, the chosen collection of topics is a good choice for a review. However, for example, the issue of boundary terms is totally neglected.

And most importantly, nothing is said about how to deal with the Ostrogradsky ghosts which must be there due to the higher derivative nature of the model. The Author correctly explains that the Faddeev-Popov ghosts are different from that, they are harmless. But what should we do with the real ghosts of the classical theory? The Author gives references for methods of fighting agaist them. But those are about non-local theories which are very different from higher-derivative ones with finite number of derivatives.

All in all, the basic idea of the review is very good. But in its present form, the target audience is absolutely unclear.

Author Response

Detailed reply to the referee no. 1

First, I sincerely thank the referee for his critical opinion and various useful suggestions regarding the topics and the style of the present article.
I improved the presentation and corrected the typos that I caught during the final reading of the manuscript.
I also changed the structure of the article a bit to avoid too long sections and subsections. I hope that adding more new sections and subsections with more precise titles helps in the comprehension of the whole article and also of the flow between various parts of it.

For the most important thing, I fully agree with the scientific referee, that the issue of unitarity (Ostrogradsky theorem or presence of "bad ghosts") should have been discussed in the realm of higher-derivative theories, which conformal gravity is an example of. I apologize for my ommission of this important problem and the lack of necessary and relevant citations to works in which various authors attempt to solve this issue with ghosts, in the previous version of the article. In order to improve the article, in the present version I added two paragraphs on the page 18 of the manuscript to discuss some of these issues and provide a list (unfortunately not an exhaustive one) of possible resolutions to this significant unitarity problem of conformal gravity, or higher-derivative theories in general. (This new part of the manuscript is now in red color to facilitate the main changes I have done on the manuscript.) I express the view, that now in the literature of the topic there are many proposed solutions to the unitarity question, but a merger of them can arise in future and may solve conclusively this only theoretical issue. I believe the conformal quantum gravity is perfectly unitary theory, now we perhaps only miss some details of the theoretical explanation of this fact of Nature.

This article has inherently a non-technical nature: the paper is more oriented to explain the main ideas to the reader, rather than showing technical details and overwhelming her/him with a lot of formulas. This is a rationale behind my choice of the scarcity of formulas. Instead I planned to describe the main ideas of covariant Faddeev-Ppopv quantization of gauge systems in words, and put formulas only there where it was really necessary. I guess such a decision does not cut out the potential audience of my paper, contrary to the statement ascribed to Hawking that every formula cuts by half number of potential readers. Therefore, I hope that even graduate students who have learned basics of QFT and covariant quantization of gauge theories can be the readers of this review.

A minor thing that the referee was kind to mention: I agree that the detailed steps in the formula (now (43)) are not needed. I just left the first and the last equality there. I only wanted to show in this formula, the results after integration by parts and with the implemented definition of the GR-covariant square operator. I agreed here and simplified my formula.

The other issue was about the boundary terms for the conformal gravity. After the suggestion of the referee, I decided to include short comment on this interesting problem on page 27. I relate the two equivalent forms of writing the action for conformal gravity. However, from the theoretical ability to extract asymptotic charges related to the generators of gauge transformations, only the second form is preferred and for this form the presence of boundary terms is crucial. This part of the new text is also in red color.

I hope that with these mentioned above changes and improvements, the paper will be ready for the publication in the Universe journal.
Once again I am grateful to the referee for his comments and suggestions on scientific additions which I hope improve and extend the content of my review.

Reviewer 2 Report

This is a review paper on methods of quantization conformal gravity.
Conformal (or Weyl) gravity is a theory of gravitation which is additionally invariant under Weyl transformation. It is described by a higher derivative action (the square of the Weyl tensor) and it does not contain any scale in its action.

It is a very interesting theory, rich of symmetries, and its quantization is (obviously) absolutely non trivial.
The author of the paper is an expert in this field, and in this paper he provides a very nice introduction and review to the problem. I have appreciated the non-technical nature of the manuscript: the paper is more oriented to explain the main ideas to the reader, rather than showing technical details.
It is a nice and very well written paper, and I certainly recommend its publication.

Author Response

I thank the referee for his favorable opinion on my contribution. I appreciate his kind words. I hope the paper will be published soon.

Reviewer 3 Report

This article is an introduction to the issue of consistently quantizing of conformal approaches to gravity. The author, among other topics, discusses the role of conformal symmetry as a gauge symmetry, presents quantization procedure preserving conformal symmetry using the Faddeev-Popov method for theories with diffeomorphisms. Also, they discuss conformal gauge-fixing, Veltman discontinuities and counting degrees of freedom in such theories. The article is extense but quite clear, and it certainly deserves publication after some minor suggestion on structure arrangement. Due to the large extension of the paper, it is sometimes difficult to follow the flow of reading, and the introduction itself looks like some kind of short review instead of only a summary of motivations and enumeration of the content of the review. The structure of only two sections for a more than 40 pages paper makes it feel overloaded for a reader who wants to get acquainted for the first time to the topic, and one suggestion would be to split the topics even more in more sections. 
Two pages after Eq.(24) (on page 29, lines 1316 and those that follow) the author mentions a discontinuity in the number of degrees of freedom for the special case of theories with $\alpha_{R^2}=0$, and mentions that due to additional constraints and symmetries this discontinuity is manifested. At this part, and probably the preceding counting of degrees of freedom could benefit from exposing, or at least mentioning how many new constraints appear and how many become second class.

Finally, I have noticed some minor typos throughout the text that should be checked before publication.

Author Response

Detailed reply to the referee no. 3

I thank the referee for his kind opinions and useful suggestions.
I changed the structure of the article a bit introducing new subsections with more detailed titles.
I also split the introduction and made it shorter and relegated some parts of it to other sections of the article. 

As for the scientific suggestion, I fully agree with the referee that it was beneficial to include also some discussion of the number of constraints and their character (1st or 2nd class) in conformal gravity. For this purpose, I added now two paragraphs, just after the mentioned formula. In these paragraphs, I intend to discuss this issue with constraints and shortly the output of the canonical quantization procedure for counting the active degrees of freedom. This part is now in red color for the easier location of changes done on the manuscript. For this part I also added some relevant citations.

Some noticed typos were corrected.

I once again thank the referee for his gentle suggestions for improvements and for his comments.

I hope that with the above changes the paper is now ready for publication.

Round 2

Reviewer 1 Report

I am still not sure how helpful can it be, to study a field by reading a text about technical issues in a non-technical style. At the same time, the topic is quite complicated in itself, so that such a review would be very nice to have. And the Author has improved the presentation indeed. I recommend to publish the paper.